# Improving Your Model Ranking on Chatbot Arena by Vote Rigging

**Rui Min** [* 1 2]  **Tianyu Pang** [* 1]  **Chao Du** [1]  **Qian Liu** [1]  **Minhao Cheng** [3]  **Min Lin** [1]

## Abstract

Chatbot Arena is a popular platform for evaluating LLMs by pairwise battles, where users vote for their preferred response from two randomly sampled anonymous models. While Chatbot Arena is widely regarded as a reliable LLM ranking leaderboard, we show that crowdsourced voting can be *rigged* to improve (or decrease) the ranking of a target model $m_t$. We first introduce a straightforward **target-only rigging** strategy that focuses on new battles involving $m_t$, identifying it via watermarking or a binary classifier, and exclusively voting for $m_t$ wins. However, this strategy is practically inefficient because there are over 190 models on Chatbot Arena and on average only about 1% of new battles will involve $m_t$. To overcome this, we propose **omnipresent rigging** strategies, exploiting the Elo rating mechanism of Chatbot Arena that any new vote on a battle can influence the ranking of the target model $m_t$, even if $m_t$ is not directly involved in the battle. We conduct experiments on around 1.7 *million* historical votes from the Chatbot Arena Notebook, showing that omnipresent rigging strategies can improve model rankings by rigging only *hundreds of* new votes. While we have evaluated several defense mechanisms, our findings highlight the importance of continued efforts to prevent vote rigging. **Code** is publicly available to reproduce all experiments.

## 1. Introduction

A variety of large language models (LLMs), both closed-source and open-source (OpenAI, 2024; Dubey et al., 2024), are now available to the community. Evaluating their alignment with human preferences is crucial for selecting suitable

---
[*]Equal contribution. The project was done during Rui Min's internship at Sea AI Lab. [1]Sea AI Lab [2]Hong Kong University of Science and Technology [3]Pennsylvania State University. Correspondence to: Tianyu Pang <tianyupang@sea.com>, Minhao Cheng <mmc7149@psu.edu>.

*Proceedings of the 42$^{nd}$ International Conference on Machine Learning*, Vancouver, Canada. PMLR 267, 2025. Copyright 2025 by the author(s).

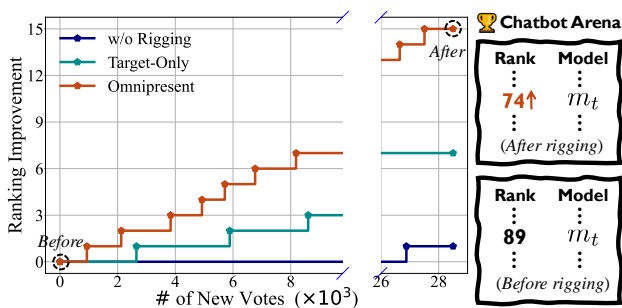

*Figure 1.* We simulate rigging on new votes beyond the $\sim 1.7$ *million* historical votes from the Chatbot Arena Notebook. In this demo, we set the target model $m_t$ as Phi-3-small-8k-Instruct. Under the normal voting distribution (w/o rigging), the ranking remains steady, showing only a single rank increase despite the addition of approximately 27,000 new votes. In contrast, vote rigging significantly boosts $m_t$'s ranking. Using the omnipresent strategy, we achieve a 15-rank improvement while being more efficient compared to the target-only strategy.

models in downstream applications (Ouyang et al., 2022). To meet this need, *Chatbot Arena* (Zheng et al., 2023a; Chiang et al., 2024) provides an open platform for conducting pairwise battles between LLMs, where users vote for their preferred response from two randomly selected anonymous models. These votes are used to compute Elo ratings for LLMs, with higher rankings on Chatbot Arena's leaderboard offering substantial promotional benefits.

Chatbot Arena is widely popular, but it relies on millions of user votes collected in the wild, which can be noisy and biased. Several strategies have been implemented to enhance the leaderboard's reliability and reduce potential gameability, including controlling for output length and style (Dubois et al., 2024; Li et al., 2024a), detecting anomalous voting patterns and bot activity (Chiang et al., 2024), categorizing prompts for data curation (Li et al., 2024b;c), and invalidating votes if anonymous model identities are revealed in the responses (Chiang et al., 2024).

Although these strategies have significantly reduced (mostly unintentional) voting biases and noise, this paper demonstrates that crowdsourced votes in Chatbot Arena can still be maliciously *rigged* to manipulate the ranking of a target model $m_t$, either improving or decreasing it. We first introduce a straightforward **target-only rigging** strategy

that focuses solely on new battles involving $m_t$, identifying it via watermarking (Zhao et al., 2024) or a binary classifier (Huang et al., 2025), and exclusively voting for $m_t$ wins. However, this strategy is practically inefficient because there are over 190 models on Chatbot Arena and on average only about $1\%$ of new battles will involve $m_t$. Improving a single ranking position for a target model may require more than 10,000 rigged votes or interactions on Chatbot Arena, a scenario that could be effectively mitigated by imposing daily voting limits per IP address (Chiang et al., 2024).

To solve this inefficiency, we observe that the Elo rating system in Chatbot Arena calculates Bradley-Terry (BT) scores (Bradley & Terry, 1952) by fitting pairwise logistic relationships on all collected votes (Eq. (1)). This implies that when a sufficient number of votes have been collected, all models' BT scores become mutually connected through these pairwise logistic relationships. Consequently, *any new vote for a battle can influence the ranking of a target model $m_t$, even if $m_t$ is not directly involved in the battle*. Based on this observation, we propose **omnipresent rigging** strategies, which first de-anonymize all models by a multi-class classifier and actively manipulate every new vote, regardless of whether $m_t$ is involved in the battle.

To prevent contaminating the actual voting records on the Chatbot Arena platform, we establish a reproducible voting environment using the publicly available historical votes from Chatbot Arena Notebook. This dataset contains around 1.7 million voting records across 129 models. In the experiments, we thoroughly examine voting scenarios under various threat models, and empirical results show that our omnipresent rigging strategies can improve model rankings by manipulating only *hundreds of* new votes. These omnipresent strategies are far more efficient than target-only strategies and other baselines, as illustrated in Figure 1. Additionally, we evaluate several defense mechanisms; however, our findings underscore the need for ongoing efforts to develop stronger protections against vote rigging.

## 2. Preliminaries

We first formalize the basic operations of Chatbot Arena in Section 2.1, including the mechanism for collecting pairwise human-annotated votes and calculating rating scores. Next, in Section 2.2, we introduce various threat models of vote rigging based on the *adversary's accessibility*.

### 2.1. Chatbot Arena

The Chatbot Arena leaderboard comprises $K$ models, denoted as $\{m_1, \ldots, m_K\}$, with their rating scores calculated on a collection of user votes $\mathbb{V}$. To collect a **new vote**, a pair of model indices $a$ and $b$ is sampled from the joint distribution $\mathcal{P}_\mathbb{V}$, where the subscript $\mathbb{V}$ indicates that the distribution

depends on previously collected votes. The user can query both sampled models $m_a$ and $m_b$ with any prompt string $s \in \mathbb{S}$, where $\mathbb{S}$ denotes the natural language space, and cast a vote for their preferred response between $m_a(s)$ and $m_b(s)$. Then the vote set $\mathbb{V}$ will be updated according to the selected voting option:

i. `a wins`: $\mathbb{V}_{a>b} = \mathbb{V} \cup \{\mathbf{e}_a - \mathbf{e}_b\}, \mathbb{V} \leftarrow \mathbb{V}_{a>b}$;

ii. `b wins`: $\mathbb{V}_{a<b} = \mathbb{V} \cup \{\mathbf{e}_b - \mathbf{e}_a\}, \mathbb{V} \leftarrow \mathbb{V}_{a<b}$;

iii. `Tie`: $\mathbb{V}_{a=b} = \mathbb{V} \cup \{\mathbf{e}_a - \mathbf{e}_b\} \cup \{\mathbf{e}_b - \mathbf{e}_a\}, \mathbb{V} \leftarrow \mathbb{V}_{a=b}$;

iv. `Abstain`: $\mathbb{V}$ is unchanged,

where $\mathbf{e}_k \in \mathbb{R}^K$ is the $k$-th basis unit vector and we slightly abuse the notation of $\cup$ to denote the appending operation.[1]

**Calculation of rating scores.** Chatbot Arena applies the Elo rating system to benchmark models. According to Chiang et al. (2024), Chatbot Arena initially used *online Elo scores* to calculate model ratings, but later switched to *Bradley-Terry (BT) scores* (Bradley & Terry, 1952) for better statistical estimation. Given a collected vote set $\mathbb{V}$, we can calculate BT scores for the $K$ models on the leaderboard, denoted in a vectorized form as $\mathbf{r}_\mathbb{V}^{\text{BT}} \in \mathbb{R}^K$, where $\mathbf{r}_\mathbb{V}^{\text{BT}}[k]$ is the BT score of the $k$-th model. The BT scores are derived from fitting the logistic relationships on $\mathbb{V}$, formulated as

$$\mathbf{r}_\mathbb{V}^{\text{BT}} = \arg\min_{\mathbf{r}} \mathbb{E}_{\mathbf{v} \in \mathbb{V}}[\mathcal{L}_{\text{BCE}}(\mathbf{v}, \mathbf{r})], \qquad (1)$$

where $\mathcal{L}_{\text{BCE}}(\mathbf{v}, \mathbf{r}) = -\log(\sigma(\mathbf{v}^\top \mathbf{r}))$ is the binary cross-entropy (BCE) loss, and $\sigma(\cdot)$ is the Sigmoid function.

### 2.2. Threat Model

Throughout this paper, our adversarial rigging goal is to **promote the ranking of a target model $m_t$** on Chatbot Arena through vote rigging. This is achieved by submitting new votes, where each voting option is strategically selected to promote the target model's ranking.

Based on the *adversary's accessibility*, we pinpoint the key elements in our threat model as described below:

- **Historical votes** ($\mathbb{V}_H$ **or** $\mathbf{r}_{\mathbb{V}_H}^{\text{BT}}$): whether the adversary has access to the historical voting data $\mathbb{V}_H$ or can only access to the BT scores $\mathbf{r}_{\mathbb{V}_H}^{\text{BT}}$ from the public leaderboard;

- **Model identities** (`Real-name` **or** `Anonymous`): whether the adversary can directly access the identities of the sampled models $m_a$ and $m_b$ in each new battle;

- **Sampling distribution** ($\mathcal{P}_\mathbb{V}$ **or** `Unknown`): whether the adversary can know the sampling distribution $\mathcal{P}_\mathbb{V}$ or not;

- **Other users' votes** ($\emptyset$ **or** $\mathbb{V}_O$): when the adversary submits (malicious) new votes, other users may also submit new votes simultaneously, denoted as $\mathbb{V}_O$.

---

[1]In our notation, we use $m_k$ and its index $k$ interchangeably to refer to the $k$-th model without causing ambiguity.

For example, when the adversary aims to manipulate the real-world Chatbot Arena platform, the threat model can be written as $\{\mathbf{r}_{\mathbb{V}_H}^{\text{BT}}, \texttt{Anonymous}, \texttt{Unknown}, \mathbb{V}_O\}$.

**Remark.** We initially contacted Chatbot Arena and disclosed the potential threat in September 2024. In our experiments, to avoid contaminating the actual voting records on the Chatbot Arena platform, we set up a reproducible voting environment using the latest historical votes (as of January 2025) that are publicly available in the Chatbot Arena Notebook. This dataset contains around 1.7 million voting records across 129 models. Within this environment, we divide 90% of the complete historical vote records as $\mathbb{V}_H$ and the remainder as $\mathbb{V}_O$ throughout all simulations.

# 3. Vote Rigging Strategies

In this section, we discuss various vote-rigging strategies aimed at promoting the ranking of the target model $m_t$. Generally, under a given threat model (where model identities are `Anonymous`), a rigging strategy manipulates *new votes* and consists of two key components:

- **The de-anonymizing function** $\mathcal{A}(s, m_k(s)) = \widetilde{k}$ takes the user prompt $s$ and the model response $m_k(s)$ as inputs, aiming to de-anonymize the true identity of $m_k$ (or its index $k$) through the predicted identity $m_{\widetilde{k}}$ (or the index $\widetilde{k}$). This function is typically trained or designed to maximize the probability $P(\widetilde{k} = k)$;

- For each new vote between the sampled models $m_a$ and $m_b$, **the vote manipulation function** $\mathcal{M}(\widetilde{a}, \widetilde{b})$ takes the identities $\widetilde{a}$ and $\widetilde{b}$ predicted by $\mathcal{A}$ as inputs and returns one of four voting options: $\widetilde{a}/a$ `wins`, $\widetilde{b}/b$ `wins`, `Tie`, or `Abstain`. Note that $\mathcal{M}$ may also depend on additional information, such as historical votes or ranks, as described in our omnipresent rigging strategy.

In the following, we elaborate on a vanilla *target-only rigging* strategy and our proposed *omnipresent rigging* strategy.

## 3.1. Target-Only Rigging

To promote the ranking of the target model $m_t$, a straightforward approach is to rig votes only for new battles predicted to involve $m_t$ (specifically, when $t \in \{\widetilde{a}, \widetilde{b}\}$). In this case, the de-anonymizing function focuses exclusively on identifying $m_t$, formulated as $\mathcal{A}_{\text{t-only}}(s, m_k(s)) \in \{t, \neg t\}$, where $\neg t$ represents all other model indices.

Two concurrent works (Zhao et al., 2024; Huang et al., 2025) have explored similar target-only rigging strategies. These works implement the de-anonymizing function $\mathcal{A}_{\text{t-only}}$ using either watermarking/attribution techniques or a binary classifier. Based on the implemented $\mathcal{A}_{\text{t-only}}$, they further

define the vote manipulation function $\mathcal{M}_{\text{t-only}}$ as

$$\mathcal{M}_{\text{t-only}}(\widetilde{a}, \widetilde{b}) = \begin{cases} a \text{ wins} & \text{if } \widetilde{a} = t, \\ b \text{ wins} & \text{if } \widetilde{b} = t, \\ \texttt{Passive} & \text{otherwise,} \end{cases} \quad (2)$$

where the `Passive` option can be set to `Tie` (T-Tie), `Abstain` (T-Abstain), a random selection (T-Random), or aligned with the normal user voting distribution (T-Normal). In our following experiments, we treat these **target-only rigging strategies as our baselines**.

## 3.2. Omnipresent Rigging

While target-only rigging strategies are straightforward, they are *inefficient* in practice, as they manipulate only the new votes predicted to involve $m_t$. For example, with over 190 models on the Chatbot Arena platform and a uniform model sampling distribution, the probability of a specific target model being involved in a battle is only about 1%. Consequently, target-only rigging strategies may *passively* select the voting options for approximately 99% of new battles. As reported in Huang et al. (2025), improving a single ranking position for a target model (e.g., from rank 129 to 128 or rank 5 to 4) requires over 10,000 votes for low-ranked models and more than 20,000 votes for high-ranked models.[2]

To enhance rigging efficiency, we draw inspiration from the following observation on Chatbot Arena's rating mechanism (an informal proof is provided in Appendix A.1):

> **Observation (omni-property):** when the BT scores $\mathbf{r}_{\mathbb{V}}^{\text{BT}}$ are calculated on a sufficient number of votes in $\mathbb{V}$ (by Eq. (1)), *any new vote on a battle between $m_a$ and $m_b$ can influence the ranking of the target model $m_t$*, even if $m_t$ is not directly involved in the battle (i.e., $t \notin \{a, b\}$).

Based on this, we propose **omnipresent rigging strategies**, which actively manipulate every new vote, regardless of whether $m_t$ is involved in the battle. We implement the de-anonymizing function $\mathcal{A}_{\text{omni}}(s, m_k(s)) \in \{1, \dots, K\}$ as a *multi-class* classifier (detailed in Appendix B). For the vote on each new battle, $\mathcal{A}_{\text{omni}}$ predicts the identities of the sampled models as $\widetilde{a}$ and $\widetilde{b}$. The design of the vote manipulation function $\mathcal{M}_{\text{omni}}$ then depends on the adversary's accessibility to historical votes, as described below.

**BT-based omni rigging (Omni-BT).** When the adversary has direct access to the historical voting data $\mathbb{V}_H$, it can combine its manipulated votes $\mathbb{V}_{\mathcal{M}}$ to form $\mathbb{V} = \mathbb{V}_H \cup \mathbb{V}_{\mathcal{M}}$. For a new battle between $m_a$ and $m_b$, the Omni-BT

---

[2]Our measure of "votes" corresponds to the "interactions" defined in Huang et al. (2025), where they use "votes" to count only the battles predicted to involve $m_t$.

manipulation function can be expressed compactly as:

$$\mathcal{M}_{\text{omni}}^{\text{BT}} = \arg\max_{\mathbb{V}'} \mathcal{R}^{\text{BT}}(\mathbf{r}_{\mathbb{V}'}^{\text{BT}}), \quad (3)$$

where $\mathbb{V}' \in \{\mathbb{V}_{\widetilde{a}<\widetilde{b}}, \mathbb{V}_{\widetilde{a}>\widetilde{b}}, \mathbb{V}_{\widetilde{a}=\widetilde{b}}, \mathbb{V}\}$ represents the four voting options: $\widetilde{a}/a$ wins, $\widetilde{b}/b$ wins, Tie, and Abstain, as introduced in Section 2.1. Here, $\mathcal{R}^{\text{BT}}(\cdot)$ denotes the **rigging objective** of Omni-BT. Throughout our experiments, we adopt the relative rating increase between $m_t$ and $m_{\widehat{t}}$ that ranks one position ahead of it with $\mathcal{R}^{\text{BT}}(\mathbf{r}_{\mathbb{V}'}^{\text{BT}}) = \mathbf{r}_{\mathbb{V}'}^{\text{BT}}[t] - \mathbf{r}_{\mathbb{V}'}^{\text{BT}}[\widehat{t}]$ as our rigging objective. We identify it through ablation studies and defer detailed comparisons in Appendix A.2. Note that since the adversary selects voting options based on the predicted $\widetilde{a}$ and $\widetilde{b}$, these predictions may deviate from the ground truth updates $\{\mathbf{r}_{\mathbb{V}_{a>b}}^{\text{BT}}, \mathbf{r}_{\mathbb{V}_{a<b}}^{\text{BT}}, \mathbf{r}_{\mathbb{V}_{a=b}}^{\text{BT}}, \mathbf{r}_{\mathbb{V}}^{\text{BT}}\}$.

**Online-based omni rigging (Omni-On).** When the adversary has access only to the up-to-date BT scores $\mathbf{r}_{\mathbb{V}_H}^{\text{BT}}$ from the public leaderboard and not to $\mathbb{V}_H$, directly optimizing Eq. (3) in Omni-BT becomes intractable. To address this, we propose using *online Elo scores* (Elo, 1967) to approximate updates to the BT scores. This approach relies exclusively on $\mathbf{r}_{\mathbb{V}_H}^{\text{BT}}$, eliminating the need for access to $\mathbb{V}_H$. Formally, after a new battle between $m_a$ and $m_b$, the online Elo scores for $m_a$ and $m_b$ are calculated as

$\mathbf{r}_a^{\text{On}}(\gamma, \mu) = \mathbf{r}_{\mathbb{V}_H}^{\text{BT}}[a] + \mu \cdot (\gamma - \mathcal{W}(\mathbf{r}_{\mathbb{V}_H}^{\text{BT}}[a], \mathbf{r}_{\mathbb{V}_H}^{\text{BT}}[b]));$
$\mathbf{r}_b^{\text{On}}(\gamma, \mu) = \mathbf{r}_{\mathbb{V}_H}^{\text{BT}}[b] + \mu \cdot (1 - \gamma - \mathcal{W}(\mathbf{r}_{\mathbb{V}_H}^{\text{BT}}[b], \mathbf{r}_{\mathbb{V}_H}^{\text{BT}}[a])),$

where $\mu$ is the step size, $\mathcal{W}(x, y) = \left(1 + 10^{(y-x)/400}\right)^{-1}$ is a logistic function, and the base 10 and scaling factor 400 are adopted following Zheng et al. (2023b). The parameter $\gamma$ depends on the voting option: $\gamma = 1$ for $a$ wins, $\gamma = 0$ for $b$ wins, and $\gamma = 0.5$ for Tie. When selecting the Abstain option, there is $\mu = 0$.

Then the Omni-On manipulation function selects the voting option for the battle between $m_a$ and $m_b$ as:

$$\mathcal{M}_{\text{omni}}^{\text{On}} = \arg\max_{\gamma, \mu} \mathcal{R}^{\text{On}}\left(\mathbf{r}_{\mathbb{V}_H}^{\text{BT}}[t], \mathbf{r}_a^{\text{On}}(\gamma, \mu), \mathbf{r}_b^{\text{On}}(\gamma, \mu)\right), \quad (4)$$

where $\gamma, \mu$ are constrained to the values corresponding to the four voting options described above, and $\mathcal{R}^{\text{On}}(\cdot)$ represents the rigging objective of Omni-On. A simple design for $\mathcal{R}^{\text{On}}(\cdot)$ is a differentiable surrogate of the ranking function. Specifically, the ranking of $m_t$ is calculated as $\text{Rank}(m_t) = 1 + \sum_{\forall k \neq t}[\mathbb{I}(\mathbf{r}_{\mathbb{V}_H}^{\text{BT}}[k] > \mathbf{r}_{\mathbb{V}_H}^{\text{BT}}[t])]$, where $\mathbb{I}(\cdot)$ is the indicator function. This ranking function can be reformulated as $\text{Rank}(m_t) = 1 + \sum_{\forall k \neq t}[\mathbb{I}(\mathcal{W}\left(\mathbf{r}_{\mathbb{V}_H}^{\text{BT}}[t], \mathbf{r}_{\mathbb{V}_H}^{\text{BT}}[k]\right) < 0.5)]$, from which we can define $\mathcal{R}^{\text{On}}$ in terms of $\mathcal{W}$, capturing the pairwise win rates. Consequently, $\mathcal{M}_{\text{omni}}^{\text{On}}$ can be defined as

$$\begin{aligned} \mathcal{M}_{\text{omni}}^{\text{On}} = \arg\max_{\gamma, \mu} & \; \mathcal{W}\left(\mathbf{r}_{\mathbb{V}_H}^{\text{BT}}[t], \mathbf{r}_a^{\text{On}}(\gamma, \mu)\right) \\ & + \mathcal{W}\left(\mathbf{r}_{\mathbb{V}_H}^{\text{BT}}[t], \mathbf{r}_b^{\text{On}}(\gamma, \mu)\right). \end{aligned} \quad (5)$$

In our experiments, we perform additional ablation studies on alternative design choices for $\mathcal{R}^{\text{On}}$ in Appendix A.3. After each vote manipulation, we can optionally update the local BT scores as $\mathbf{r}_{\mathbb{V}_H}^{\text{BT}}[a] \leftarrow \mathbf{r}_a^{\text{On}}(\gamma, \mu)$ and $\mathbf{r}_{\mathbb{V}_H}^{\text{BT}}[b] \leftarrow \mathbf{r}_b^{\text{On}}(\gamma, \mu)$. However, our empirical results in Appendix A.4 show that it would be better to keep $\mathbf{r}_{\mathbb{V}_H}^{\text{BT}}$ unchanged during each Omni-On manipulation.

# 4. Sanity Check with Idealized Rigging

We start by rigging against the idealized scenario using the threat model $\{\mathbb{V}_H, \text{Real-name}, \mathcal{P}_{\mathbb{V}}, \emptyset\}$. Results from this sanity check indicate our optimal rigging performance and serve as an *upper bound* for the capability analysis in Section 5. Without specific assumptions on the sampling distribution $\mathcal{P}_{\mathbb{V}}$, we use uniform sampling with the marginal probability of a sampling $m_k$ being $P_k = \frac{2}{K}$. Additionally, to understand how effective vote rigging performs, we include the w/o rigging case in which votes are sampled using the normal user voting distribution as comparisons. We report our initial results by rigging 20,000 new votes and defer results with larger numbers of votes to Appendix C.1. We demonstrate the ranking changes of diverse target models $m_t$ including Llama-2-13B-Chat (Touvron et al., 2023), Mistral-7B-Instruct-v0.2 (Jiang et al., 2023), Qwen1.5-14B-Chat (Bai et al., 2023), Vicuna-7B (Chiang et al., 2023b), Gemma-2-9B-it (Gemma et al., 2024b), and Phi-3-small-8k-Instruct (Abdin et al., 2024) and defer rigging results with 22 extra models (used in Huang et al. (2025)) to Appendix C.2.

As shown in Figure 2, *all rigging strategies effectively improve $m_t$'s ranking* compared to the scenarios without rigging, achieving an average of 6-rank improvement. Besides, our omnipresent strategies demonstrate *significantly higher rigging efficiency* against target-only strategies. For example, when rigging 20,000 new votes, the target-only rigging achieves only an average increase of 4, whereas both omnipresent rigging strategies notably outperform it, resulting in an approximately 10-rank promotion.

# 5. On Exploring the Rigging Capability

However, practical vote rigging is typically conducted with limited *adversary's accessibility*, potentially reducing the manipulation effectiveness. Here, we conduct a series of stress tests to explore whether our strategies remain effective against these more demanding rigging scenarios. Specifically, in Section 5.1 we conduct rigging under the threat model $\{\mathbb{V}_H, \text{Anonymous}, \mathcal{P}_{\mathbb{V}}, \emptyset\}$ to explore the impact of inaccurate de-anonymization with predicted probability $P(\widetilde{k} = k) < 1$, then in Section 5.2 we conduct rigging under the threat model $\{\mathbb{V}_H, \text{Real-name}, \text{Unknown}, \emptyset\}$ to simulate the influence of Unknown sampling distribution, and finally in Section 5.3, we conduct rigging under the

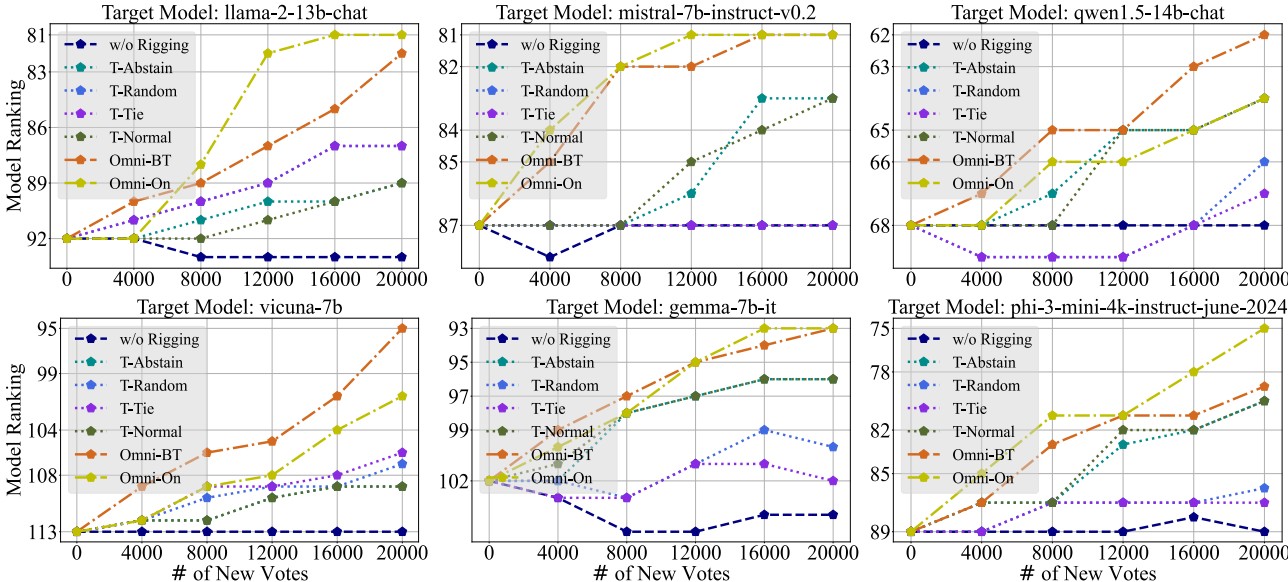

*Figure 2.* Ranking improvements under the idealized rigging scenario with different target models $m_t$. Our omnipresent rigging strategies (Omni-BT and Omni-On) result in approximately double the ranking promotions compared to target-only strategies.

threat model $\{\mathbb{V}_H, \texttt{Real-name}, \mathcal{P}_\mathbb{V}, \mathbb{V}_O\}$ to incorporate the influence of concurrent user voting.

### 5.1. Rigging with Inaccurate De-Anonymization

Since our rigging strategies rely on $m_{\widetilde{k}}$ predicted by the de-anonymizing function $\mathcal{A}(\cdot)$ to select voting options, its predicted probability $P(\widetilde{k} = k)$ thus directly impacts the rigging effectiveness. To examine whether vote rigging remains effective against inaccurate de-anonymization, i.e., $P(\widetilde{k} = k) < 1$, we set a proportion of battles to `Anonymous` model identities. As shown in Table 1, all rigging strategies exhibit decreased ranking promotion as expected, with both T-Tie and T-Random achieving no manipulation effect when half of the battles have `Anonymous` model identities. Besides, we observe that the Omni-On exhibits more resistance against inaccurate de-anonymization than other strategies. These results can be attributed to the usage of initial $\mathbf{r}_{\mathbb{V}_H}^{\text{BT}}$ for Omni-On without updating with manipulated votes $\mathbb{V}_\mathcal{M}$, which makes it more resistant to the impact of previously misclassified model identities.

### 5.2. Rigging with Unknown Sampling Distribution

Practical sampling distributions could be `Unknown` to users, for example, newly released models might acquire a higher sampling probability to collect enough votes (Zhao et al., 2024). As a result, these non-uniform sampling strategies might potentially reduce $P_t$, i.e., the marginal probability of sampling $m_t$, thereby decreasing the number of sampled battles containing $m_t$. In this section, we sample new battles using $P_t = \beta \cdot \frac{2}{K}$, where $\beta \in [0, 1]$ controls the degree of probability reduction. When $\beta = 0$, it indicates that

no $m_t$ will be sampled for new battles. As shown in Table 2, decreasing $P_t$ significantly reduces the effectiveness of target-only rigging, with most strategies failing completely at $\beta = 0.3$. In contrast, omnipresent strategies show effective manipulation performance with over 5-rank improvement even when $m_t$ is not directly involved in battles.

### 5.3. Rigging with Concurrent User Voting

In addition to manipulated votes $\mathbb{V}_\mathcal{M}$, concurrent votes $\mathbb{V}_O$ from other users remain unknown to the adversary, which could affect the rigging effectiveness. For instance, they would lead to an inaccurate calculation of omnipresent rigging objectives $\mathcal{R}^{\text{BT}}(\cdot)$ and $\mathcal{R}^{\text{On}}(\cdot)$, thereby impacting the subsequent vote selection of their respective manipulation functions $\mathcal{M}_{\text{omni}}^{\text{BT}}(\cdot)$ and $\mathcal{M}_{\text{omni}}^{\text{On}}(\cdot)$. To incorporate the influence of $\mathbb{V}_O$, we use the combined votes $\mathbb{V} = \mathbb{V}_H \cup \mathbb{V}_\mathcal{M} \cup \mathbb{V}_O$ to calculate the final rating. Our results in Table 3 demonstrate that the influence of $\mathbb{V}_O$ remains minor, which only introduces an average 1-rank decrease even with a $\mathbb{V}_O$ containing 100,000 votes. These findings suggest the resilience of vote rigging against concurrent user voting.

## 6. Case Study: Rigging Chatbot Arena

### 6.1. Towards Simulating Real-world Vote Rigging

To demonstrate how to improve target model $m_t$'s ranking in the realistic leaderboard, we simulate vote rigging against the practical scenario with the threat model being $\{\mathbf{r}_{\mathbb{V}_H}^{\text{BT}}, \texttt{Anonymous}, \texttt{Unknown}, \mathbb{V}_O\}$. Through this case study, our preliminary findings would serve as a proof-of-concept that exposes the real-world rigging risks within the

*Table 1.* Results of rigging performance against different proportion of `Anonymous` battles.

| Method | Ranking↓ (Ranking Increase↑) | | | | |
|---|---|---|---|---|---|
| | 10% | 20% | 30% | 40% | 50% |
| T-Tie | 90 (+2) | 91 (+1) | 91 (+1) | 92 (+0) | 92 (+0) |
| T-Abstain | 87 (+5) | 88 (+4) | 89 (+3) | 89 (+3) | 90 (+2) |
| T-Random | 90 (+2) | 90 (+2) | 90 (+2) | 91 (+1) | 92 (+0) |
| T-Normal | 87 (+5) | 88 (+4) | 88 (+4) | 88 (+4) | 89 (+3) |
| Omni-BT | 84 (+8) | 84 (+8) | 86 (+6) | 87 (+5) | 88 (+4) |
| Omni-On | 84 (+8) | 84 (+8) | 85 (+7) | 86 (+6) | 86 (+6) |

*Table 2.* Vote rigging under various marginal probability $P_t = \beta \cdot \frac{2}{K}$. When $\beta = 0$, it indicates that no $m_t$ will be sampled.

| Method | Ranking↓ (Ranking Increase↑) | | | | |
|---|---|---|---|---|---|
| | $\beta = 0.0$ | $\beta = 0.3$ | $\beta = 0.5$ | $\beta = 0.7$ | $\beta = 0.9$ |
| T-Tie | 95 (-3) | 94 (-2) | 93 (-1) | 92 (+0) | 90 (+2) |
| T-Abstain | 92 (+0) | 91 (+1) | 89 (+3) | 89 (+3) | 87 (+5) |
| T-Random | 95 (-3) | 92 (+0) | 92 (+0) | 91 (+1) | 90 (+2) |
| T-Normal | 92 (+0) | 90 (+2) | 89 (+3) | 88 (+4) | 87 (+5) |
| Omni-BT | 87 (+5) | 86 (+6) | 84 (+8) | 84 (+8) | 83 (+9) |
| Omni-On | 86 (+6) | 85 (+7) | 84 (+8) | 84 (+8) | 83 (+9) |

*Table 3.* Rigging results against various scales of $\mathbb{V}_O$.

| Method | Ranking↓ (Ranking Increase↑) | | | | |
|---|---|---|---|---|---|
| | $2 \times 10^4$ | $4 \times 10^4$ | $6 \times 10^4$ | $8 \times 10^4$ | $10^5$ |
| T-Tie | 90 (+2) | 90 (+2) | 90 (+2) | 90 (+2) | 90 (+2) |
| T-Abstain | 87 (+5) | 87 (+5) | 87 (+5) | 87 (+5) | 88 (+4) |
| T-Random | 89 (+3) | 89 (+3) | 90 (+2) | 90 (+2) | 90 (+2) |
| T-Normal | 87 (+5) | 87 (+5) | 87 (+5) | 87 (+5) | 88 (+4) |
| Omni-BT | 82 (+10) | 82 (+10) | 82 (+10) | 82 (+10) | 82 (+10) |
| Omni-On | 83 (+9) | 83 (+9) | 83 (+9) | 83 (+9) | 84 (+8) |

*Table 4.* Rigging with prompts from the Quora (Q) and HC3 (H) datasets. We denote Target-Only* as the idealized T-Abstain.

| Method | Ranking↓ (Ranking Increase↑) | | | |
|---|---|---|---|---|
| | Llama | Mistral | Qwen | Vicuna |
| w/o Rigging | 15 (+1) | 13 (+0) | 12 (-1) | 21 (+0) |
| Target-Only* | 15 (+1) | 11 (+2) | 10 (+1) | 18 (+3) |
| Omni-BT (H) | 11 (+5) | 9 (+4) | 8 (+3) | 12 (+9) |
| Omni-On (H) | 14 (+2) | 10 (+3) | 9 (+2) | 17 (+4) |
| Omni-BT (Q) | 10 (+6) | 9 (+4) | 9 (+2) | 12 (+9) |
| Omni-On (Q) | 14 (+2) | 10 (+3) | 10 (+1) | 17 (+4) |

Chatbot Arena. Specifically, we extract 25 models with around 23,000 English-specific votes from the complete historical records to set up the simulation environment. We present ranking improvements of the target model $m_t$, which is set to be one of the four models including Llama-2-13B-Chat, Mistral-7B-Instruct-v0.2, Qwen1.5-14B-Chat, and Vicuna-7B. Details on the overall model selection can be found in Appendix B.2.

**Setup for target-only rigging.** We employ the idealized rigging with $P(\widetilde{t} = t) = 1$ since the de-anonymizing function $\mathcal{A}_{\text{t-only}}(\cdot)$ in Zhao et al. (2024); Huang et al. (2025) typically achieves a high prediction performance. Besides, we adopt T-Abstain as it delivers a more stable ranking improvement.

**Setup for omnipresent rigging.** We fine-tune RoBERTa-based classifiers (Liu et al., 2019) to classify all 25 models with two datasets respectively, including the HC3 and Quora datasets. We generate the training corpus by querying each model with 4,000 training prompts. The fine-tuning process includes 20 epochs with a batch size of 64 which takes a few hours on 2× NVIDIA A100 GPUs. When simulating the vote rigging, we reuse the training prompts to query model pairs within each sampled battle. We defer more details of the training corpus to Appendix B.3.

**Vote rigging can be effective in practice.** Table 4 demonstrates that both of our omnipresent strategies outperform the target-only strategy. Our Omni-BT and Omni-On strategies achieve an average of 5-rank and 3-rank promotion, respectively, yielding more than 50% ranking improvement

compared to the optimal performance of T-Abstain rigging.

### 6.2. Ablation Studies on Omnipresent Rigging

**What if unseen prompts are employed for vote rigging?** While our initial results use training prompts for rigging, reusing these limited prompts could be easily detected by the quality control in Chatbot Arena such as the simple prompt-deduplication strategy (Chiang et al., 2024). As a result, we aim to investigate whether unseen prompts are effective for rigging, especially without classifier retraining. Our results in Figure 3 (a) show that both omnipresent strategies still effective ranking improvement even rigging with unseen prompts. These preliminary findings indicate the potential scalability of using the multi-class classifier for de-anonymizing rather than directly relying upon the memorization of trained responses.

**Explore the effectiveness of omni rigging with unrecognized models.** Since the Chatbot Arena may constantly introduce new models to the leaderboard, which become unrecognized by our trained classifier. To investigate how these models will affect the rigging performance, we include 5 additional models (described in Appendix B.2) that are outside the classification range and conduct rigging within these 30 models. Figure 3 (b) shows that both omnipresent strategies outperform target-only rigging, demonstrating a degree of resilience against unrecognized models.

**Rigging the length-control leaderboard.** In addition to the original leaderboard, the Chatbot Arena offers the length-

*Table 5.* Rigging results against the length-control leaderboard.

| Method | Ranking↓ (Ranking Increase↑) | | | |
|---|---|---|---|---|
| | Llama | Mistral | Qwen | Vicuna |
| Omni-BT (H) | 13 (+3) | 11 (+2) | 9 (+2) | 8 (+13) |
| Omni-On (H) | 15 (+1) | 10 (+3) | 9 (+2) | 15 (+6) |
| Omni-BT (Q) | 13 (+3) | 11 (+2) | 9 (+2) | 10 (+11) |
| Omni-On (Q) | 14 (+2) | 11 (+2) | 9 (+2) | 10 (+11) |

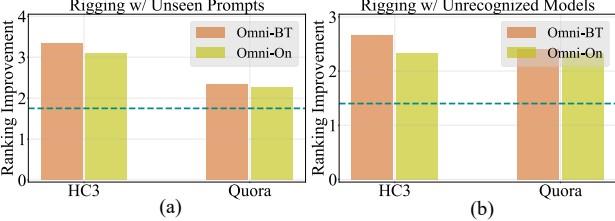

*Figure 3.* The left figure depicts the rigging results simulated with unseen prompts, while the right figure shows rigging under the impact of unrecognized models. The dashed lines represent the performance of idealized T-Abstain rigging.

*Table 6.* The proportion of correctly classified responses obtained from the realistic Chatbot Arena platform (side by side).

| Dataset | Llama3 | | GPT-4o | | Gemma | |
|---|---|---|---|---|---|---|
| | Train | Unseen | Train | Unseen | Train | Unseen |
| HC3 | 30/30 | 30/30 | 30/30 | 30/30 | 25/30 | 22/30 |
| Quora | 25/30 | 25/30 | 28/30 | 27/30 | 29/30 | 27/30 |

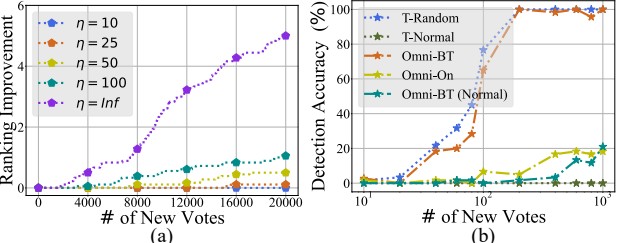

*Figure 4.* We present two strategies to detect anomalous voting behavior. The left figure detects and prevents users from submitting duplicate votes while the right figure identifies malicious voting behavior that deviates from the normal user voting distribution.

control version, which explicitly disentangles the effect of response length in rating calculation. Here we aim to investigate whether our vote rigging is still effective when conducted on the length-control leaderboard. As illustrated in Table 5, our omnipresent rigging still maintains an effective rigging effect achieving an average of 4-rank improvement. Furthermore, we observe an intriguing phenomenon in which Vicuna-7B's ranking is significantly boosted compared to rigging the original leaderboard. While our initial strategies are not specifically designed to achieve this, our findings highlight another vulnerability within the length-control mechanism, where adversaries could optimize prompts to reduce length discrepancies between responses, thereby increasing the importance of rigged votes in updating the length-control leaderboard.

**De-anonymizing responses from the Chatbot Arena.** While previous simulations use locally generated responses for de-anonymizing, it remains unclear whether our classifier performs effectively on realistic responses from Chatbot Arena. To investigate it, we use the first 30 training and unseen prompts to generate responses from three models including Llama-3-8B-Instruct (Dubey et al., 2024), GPT-4o-mini-2024-07-18 (Achiam et al., 2023), and Gemma-2-9B-it (Gemma et al., 2024b) through Chatbot Arena's APIs. We provide response examples from the HC3 in Appendix D. As shown in Table 6, our classifier still effectively distinguishes these responses, highlighting the practical effectiveness of our classifier-based de-anonymizing function.

**Efficiency analysis of classifier training.** In this section, we leverage various scales of training corpus for classifier training to demonstrate the efficiency of classifier training

in practice. For evaluation, we generate 1,000 responses for each model using unseen prompts and report their average accuracy. Results in Table 7 show that the classifier maintains comparable performance even when the training dataset is reduced to half of its original size.

## 7. Defense against Vote Rigging

To mitigate the risks of ranking manipulation, we discuss several methods to defend against vote rigging, including detecting malicious users and filtering anomalous votes.

**Detect users with duplicate vote submission.** Given that both the T-Abstain/T-Tie rigging strategies consistently vote `Abstain`/`Tie` for the `Passive` options, a straightforward defense involves detecting and preventing such duplicate voting behavior. For instance, if a continuous voting duplication is detected over $\eta$ battles, the user will be suspended from voting for a period (e.g., we discard the following 200 new votes from the violating user in our demo). A larger $\eta$ implies weaker detection but less impact on normal voting, with $\eta = \inf$ indicating no detection of duplicate votes. Our defense results against T-Abstain in Figure 4 (a) show that our simple mechanism can effectively eliminate the ranking increase by 80%, even with a large $\eta = 100$.

**Identify malicious users.** While detecting duplicate votes is effective against T-Abstain/T-Tie, practical adversaries may submit random `Passive` options (T-Random) to simply bypass the detection. To overcome the challenge, Chiang et al. (2024); Huang et al. (2025) have discussed an identification mechanism, which detects anomalous voting behavior that deviates from the normal user voting distri-

*Table 7.* The Top-1 and Top-5 accuracy against various scales of the training corpus.

| Accuracy | Training Prompts per Model | | | | |
|---|---|---|---|---|---|
| | 2000 | 2500 | 3000 | 3500 | 4000 |
| Top-1 | 76.92% | 78.34% | 79.09% | 79.96% | 79.23% |
| Top-5 | 96.34% | 97.08% | 97.28% | 97.72% | 97.32% |

*Table 8.* The mitigation results of the vote filtering strategy.

| Method | Ranking↓ (Ranking Increase↑) | | | | |
|---|---|---|---|---|---|
| | $\tau = 0.7$ | $\tau = 0.75$ | $\tau = 0.8$ | $\tau = 0.85$ | $\tau = 0.9$ |
| T-Tie | 91 (+1) | 90 (+2) | 89 (+3) | 89 (+3) | 89 (+3) |
| T-Abstain | 89 (+3) | 88 (+4) | 87 (+5) | 87 (+5) | 87 (+5) |
| T-Random | 90 (+2) | 89 (+3) | 89 (+3) | 89 (+3) | 89 (+3) |
| T-Normal | 88 (+4) | 88 (+4) | 87 (+5) | 87 (+5) | 87 (+5) |
| Omni-BT | 86 (+6) | 85 (+7) | 84 (+8) | 83 (+9) | 82 (+10) |
| Omni-On | 85 (+7) | 84 (+8) | 83 (+9) | 83 (+9) | 83 (+9) |

bution. Here we follow the implementation of Huang et al. (2025), which leverages a likelihood test against the null hypothesis where votes are from normal users. Results in Figure 4 (b) demonstrate its effectiveness in detecting the T-Random and Omni-BT rigging. However, as suggested by Huang et al. (2025), the adversary could bypass it by casting normal votes with the public ranking (T-Normal). For Omni-BT, when casting around 20% of normal votes, we successfully reduce the detection accuracy to 20% despite a less than 15% decrease in ranking promotion. Additionally, our original Omni-On is challenging to detect without adaption, indicating its stealthier rigging behavior in practice.

**Vote filtering with pairwise win rates.** Since practical adversaries may have multiple accounts to rig which reduces the effectiveness of anomaly user detections. To address this issue, we propose a simple vote filtering designed to remove anomalous votes that deviate from the historical win rate: For each collected vote, if it satisfies $\mathcal{W}(\mathbf{r}^{\text{BT}}_{\mathbb{V}_H}[a], \mathbf{r}^{\text{BT}}_{\mathbb{V}_H}[b]) > \tau$ with $m_b$ `wins` or $\mathcal{W}(\mathbf{r}^{\text{BT}}_{\mathbb{V}_H}[b], \mathbf{r}^{\text{BT}}_{\mathbb{V}_H}[a]) > \tau$ with $m_a$ `wins`, we then discard it for leaderboard updating. The $\tau$ controls the filtering proportion and the basic intuition here is to reduce unlikely voting results calibrated by pairwise win rates. We provide a detailed implementation in Appendix E. As shown in Table 8, while vote filtering reduces the overall ranking improvement, it still suffers from completely eliminating the rigging effect, where our omnipresent strategies still achieve over 6-rank improvement even with $\tau = 0.7$. In conclusion, our findings demonstrate the difficulty of thoroughly defending against vote rigging, implying that more effective defenses should be developed to improve Chatbot Arena's integrity.

# 8. Related Work

**LLM evaluation.** Developing LLMs benchmarking is a crucial task for measuring their intrinsic capabilities. Conventional benchmarks like GLUE (Wang et al., 2018), HumanEval (Chen et al., 2021), MMLU (Hendrycks et al., 2020), and GSM-8K (Cobbe et al., 2021) assess LLMs in a static manner, where they typically rely on predefined test cases. Although convenient, these benchmarks are difficult to comprehensively capture the open-ended generation capabilities (Liang et al., 2023; Peng et al., 2022) of emerging advanced models, and are typically associated

with concerns such as dataset contamination (Yang et al., 2023; Sainz et al., 2023) and Out-Of-Distribution robustness (Yuan et al., 2023). To address these challenges, recent progresses (Zheng et al., 2023a; Li et al., 2023; Dubois et al., 2024) employ LLM-as-a-Judge where a *strong* language model such as GPT-4 (Achiam et al., 2023) serves as a referee for model assessment. While reducing the need for human annotation, these automatic evaluators might suffer from spurious features, such as verbosity and position bias (Dubois et al., 2024; Chen et al., 2024; Liu et al., 2024). Unlike traditional benchmarks, Chatbot Arena (Chiang et al., 2024) devises an online platform that allows site users to vote between a pair of anonymous models based on preferred responses. By leveraging crowdsourced voting, the leaderboard aggregates high-diversity human-annotated votes, which features Chatbot Arena the most popular and widely recognized LLM benchmark.

**Vulnerability of LLM evaluation.** Previous studies (Raina et al., 2024; Shi et al., 2024; Zheng et al., 2024) have exposed the vulnerability of the LLM-as-a-Judge by adversarially cheating the LLM evaluator. While these studies primarily concentrate on identifying vulnerabilities in automatic evaluation paradigms, our paper distinguishes them with a focus on rigging the human-voted Chatbot Arena. In concurrent with our work, Zhao et al. (2024); Huang et al. (2025) leverage strategies such as watermarking and binary-classifier to identify and exclusively vote for the target model $m_t$, which can be absorbed within our general **target-only rigging** strategy. Additionally, we provide a more unified rigging framework along with an in-depth analysis of rigging capability. Our proposed **omnipresent rigging** strategy significantly improves the rigging efficiency and is effective even if $m_t$ is not directly involved in battles.

**Vulnerabilities of social voting systems.** Social voting has become an effective method for gathering crowdsourced opinions, but it is often challenged due to potential vulnerabilities (Nitzan, 1985; Islam et al., 2010; Bassi, 2015), such as strategic manipulation. Gibbard (1973) analyzes the general vulnerability of preference aggregation to voting manipulation; Hartvigsen (2008) examines vulnerabilities in political elections; Peters & Veselova (2023) discusses

the threat of group manipulation; and Wang et al. (2015); Feng et al. (2010) propose defense mechanisms to enhance the trustworthiness of voting systems. While these studies focus on various forms of strategic manipulation, our work introduces several unique challenges when manipulating the Chatbot Arena. Specifically, unlike previous manipulation settings (Bassi, 2015), where voting candidates are visible to voters, the sampled models in our case are initially *anonymous* before any rigging occurs. To address this challenge, we design and implement several effective model identification strategies, denoted as $\mathcal{A}$. Additionally, to improve the *rigging efficiency* in practice, we design two general-purpose rigging objectives ($\mathcal{R}^{\text{BT}}$ and $\mathcal{R}^{\text{On}}$), which significantly outperform baselines and the two most relevant prior works in terms of efficiency.

**LLM identification through model responses.** Identifying the source of LLM through model responses has been widely researched. Active identification methods such as LLM watermarking (Kirchenbauer et al., 2023; Yoo et al., 2023; Fernandez et al., 2023; Christ et al., 2024; Kirchenbauer et al., 2024) and LLM backdoor (Shu et al., 2023; Li et al., 2024d; Hubinger et al., 2024; Yan et al., 2024; Xu et al., 2024; Rando & Tramèr, 2023) embed traceable information into LLM responses, facilitating further detection through predefined statistical metrics. On the other hand, passive strategies (Dou et al., 2022; Guo et al., 2023; Chen et al., 2023a; Ghosal et al., 2023; Chen et al., 2023b; Verma et al., 2024) analyze hidden text style without altering the generation process. For instance, Guo et al. (2023) fine-tuned a RoBERTa-based model (Liu et al., 2019) to distinguish between output preference for LLM-generated responses and human-written documents.

## 9. Conclusion

In this paper, we expose the vulnerability within Chatbot Arena where rankings of target model $m_t$ can be improved through a simple **target-only rigging** strategy. However, given the large number of models on Chatbot Arena, this strategy could be practically inefficient. To tackle this, we further propose the **omnipresent rigging** strategy by re-designing rigging objectives with omni-property, which significantly improves the rigging efficiency and is effective even without directly rigging $m_t$. While our study primarily presents proof-of-concept experiments, practical adversaries could simply use the multi-class classifier or more advanced de-anonymizing functions $\mathcal{A}_{\text{omni}}(\cdot)$ to predict model identities and cast malicious new votes to boost $m_t$'s ranking with substantial promotional benefits. In conclusion, our findings highlight the challenges of providing a faithful LLM evaluation with human-annotated votes. Furthermore, devising effective anti-rigging defenses would be critical in future research to preserve the integrity of not only the Chatbot Arena but also emerging voting-based evaluation systems.

## Impact Statement

Due to Chatbot Arena's widespread popularity in LLM evaluation, it is possible that practical adversaries could exploit our rigging strategies to improve their own target models' ranking for substantial promotional benefits. These malicious behaviors would put other normal model developers' interests at risk, and even worse, undermine the reliability and trustworthiness of Chatbot Arena. On the other hand, while we have discussed several defense methods against vote rigging, our initial attempts at rigging defense highlight the difficulties in completely eliminating the manipulation effect. As a result, we encourage the community to focus on developing more robust defense mechanisms to mitigate the rigging vulnerabilities of Chatbot Arena as well as strengthening the integrity of emerging voting-based evaluation systems such as Copilot Arena and WebDev Arena.

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

# A. Ablation Studies of Different Rigging Objectives for Omnipresent Rigging

### A.1. An Informal Proof of Omni-Property

Given a collected voting set $\mathbb{V}$ and the target model $m_t$, we assume, without loss of generality, that a (malicious or normal) user votes for $a$ `wins` in a new battle between $m_a$ and $m_b$, where $t \notin \{a, b\}$. After this new vote, the voting set is updated to $\mathbb{V}_{a>b}$ as described in Section 2.1.

It is directly evident by Eq. (1) that the BT scores on $m_a$ and $m_b$ will change, i.e., $\mathbf{r}_{\mathbb{V}}^{\text{BT}}[a] \neq \mathbf{r}_{\mathbb{V}_{a>b}}^{\text{BT}}[a]$ and $\mathbf{r}_{\mathbb{V}}^{\text{BT}}[b] \neq \mathbf{r}_{\mathbb{V}_{a>b}}^{\text{BT}}[b]$. Then since the sampling distribution $\mathcal{P}$ is always non-zero on all battle pairs and the collected voting set $\mathbb{V}$ is assumed to be *sufficiently large*, it is reasonable to conclude that at least one vote on the battle between $m_t$ and $m_a$ or $m_t$ and $m_b$ is included in $\mathbb{V}$. Consequently, the value of $\mathbf{r}_{\mathbb{V}}^{\text{BT}}[t]$ depends on $\mathbf{r}_{\mathbb{V}}^{\text{BT}}[a]$ and/or $\mathbf{r}_{\mathbb{V}}^{\text{BT}}[b]$, and $\mathbf{r}_{\mathbb{V}_{a>b}}^{\text{BT}}[t]$ depends on $\mathbf{r}_{\mathbb{V}_{a>b}}^{\text{BT}}[a]$ and/or $\mathbf{r}_{\mathbb{V}_{a>b}}^{\text{BT}}[b]$. Thus, we can conclude that $\mathbf{r}_{\mathbb{V}}^{\text{BT}}[t] \neq \mathbf{r}_{\mathbb{V}_{a>b}}^{\text{BT}}[t]$, indicating that a new vote on the battle between $m_a$ and $m_b$ will influence the BT score of the target model $m_t$. $\qquad\square$

### A.2. Does Improving Relative Rating Increase Better than Improving Absolute Rating Increase for Omni-BT

In this section, we illustrate why we choose $\mathcal{R}^{\text{BT}}(\mathbf{r}_{\mathbb{V}'}^{\text{BT}}) = \mathbf{r}_{\mathbb{V}'}^{\text{BT}}[t] - \mathbf{r}_{\mathbb{V}'}^{\text{BT}}[\widehat{t}]$ that measures the relative rating increase between $m_t$ and $m_{\widehat{t}}$, i.e., the model that ranks one position ahead of $m_t$ as our rigging objective. For comparison, we implement a straightforward objective $\mathcal{R}^{\text{BT}}(\mathbf{r}_{\mathbb{V}'}^{\text{BT}}) = \mathbf{r}_{\mathbb{V}'}^{\text{BT}}[t]$ that directly maximizes the absolute rating increase. We reconduct experiments under the setting in Section 4 and present comparison results of their average ranking increase across all manipulated votes in Figure 5. It is observed that by maximizing the relative rating increase, we achieve a more stable and efficient ranking promotion. In practice, the adversary may explore more effective rigging objectives, which is worth discussing in future studies.

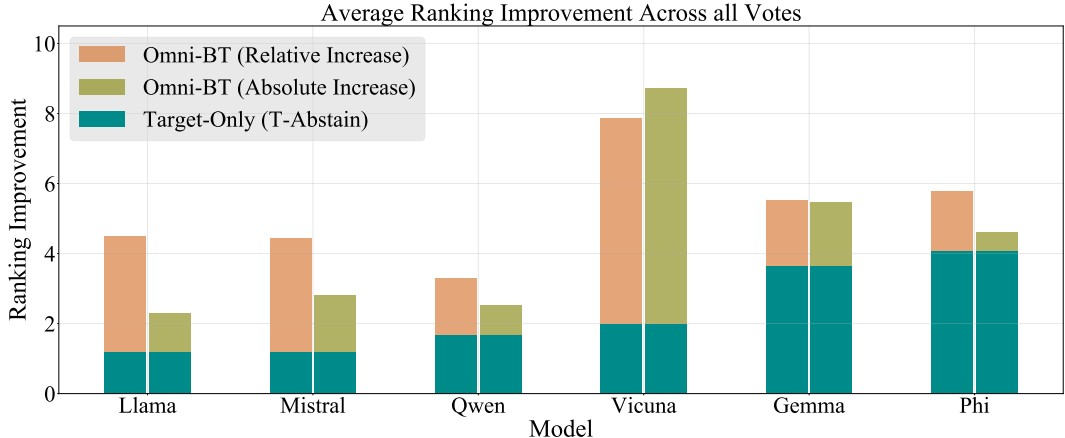

*Figure 5.* We show the average ranking improvement across all new votes for two rigging objectives, where *Relative Increase* indicates the rigging objective $\mathcal{R}^{\text{BT}}(\mathbf{r}_{\mathbb{V}'}^{\text{BT}}) = \mathbf{r}_{\mathbb{V}'}^{\text{BT}}[t] - \mathbf{r}_{\mathbb{V}'}^{\text{BT}}[\widehat{t}]$ that maximizes $m_t$'s relative rating increase between $m_t$ and $m_{\widehat{t}}$ and *Absolute Increase* indicates the rigging objective $\mathcal{R}^{\text{BT}}(\mathbf{r}_{\mathbb{V}'}^{\text{BT}}) = \mathbf{r}_{\mathbb{V}'}^{\text{BT}}[t]$ that maximizes the absolute rating increase of $m_t$.

### A.3. What if Maximizing the Win Rate of One Model for Omni-On

Our original Omni-On strategy in Eq. (5) aims to maximize the average pairwise win rates over $m_a$ and $m_b$. Here, we investigate an intriguing question: what if we only consider maximizing the win rate of one model (either $m_a$ or $m_b$)? We formulate our problem into two straightforward rigging objectives: the first involves maximizing the win rate over the model with a higher ranking, with the objective as

$$\mathcal{R}^{\text{On-Min}}\left(\mathbf{r}_{\mathbb{V}_H}^{\text{BT}}[t], \mathbf{r}_a^{\text{On}}(\gamma, \mu), \mathbf{r}_b^{\text{On}}(\gamma, \mu)\right) = \min(\mathcal{W}\left(\mathbf{r}_{\mathbb{V}_H}^{\text{BT}}[t], \mathbf{r}_a^{\text{On}}(\gamma, \mu)\right), \mathcal{W}\left(\mathbf{r}_{\mathbb{V}_H}^{\text{BT}}[t], \mathbf{r}_b^{\text{On}}(\gamma, \mu)\right)), \qquad (6)$$

and the other one focuses on maximizing the win rate over the lower-ranking models, with the following objective

$$\mathcal{R}^{\text{On-Max}}\left(\mathbf{r}_{\mathbb{V}_H}^{\text{BT}}[t], \mathbf{r}_a^{\text{On}}(\gamma, \mu), \mathbf{r}_b^{\text{On}}(\gamma, \mu)\right) = \max(\mathcal{W}\left(\mathbf{r}_{\mathbb{V}_H}^{\text{BT}}[t], \mathbf{r}_a^{\text{On}}(\gamma, \mu)\right), \mathcal{W}\left(\mathbf{r}_{\mathbb{V}_H}^{\text{BT}}[t], \mathbf{r}_b^{\text{On}}(\gamma, \mu)\right)). \qquad (7)$$

*Table 9.* Rigging results against various numbers of votes.

| Method | Vote Numbers | | | | | | | | | |
|---|---|---|---|---|---|---|---|---|---|---|
| | 5000 | 10000 | 15000 | 20000 | 25000 | 30000 | 35000 | 40000 | 45000 | 50000 |
| w/o Rigging | 92 (+0) | 92 (+0) | 92 (+0) | 92 (+0) | 92 (+0) | 92 (+0) | 92 (+0) | 92 (+0) | 92 (+0) | 92 (+0) |
| T-Tie | 92 (+0) | 91 (+1) | 90 (+2) | 90 (+2) | 89 (+3) | 88 (+4) | 86 (+6) | 83 (+9) | 82 (+10) | 79 (+13) |
| T-Abstain | 91 (+1) | 89 (+3) | 88 (+4) | 87 (+5) | 86 (+6) | 85 (+7) | 83 (+9) | 81 (+11) | 80 (+12) | 79 (+13) |
| T-Random | 92 (+0) | 90 (+2) | 90 (+2) | 89 (+3) | 88 (+4) | 86 (+6) | 84 (+8) | 83 (+9) | 80 (+12) | 79 (+13) |
| T-Normal | 91 (+1) | 89 (+3) | 88 (+4) | 87 (+5) | 86 (+6) | 86 (+6) | 84 (+8) | 82 (+10) | 81 (+11) | 80 (+12) |
| Omni-BT | 87 (+5) | 86 (+6) | 84 (+8) | 82 (+10) | 80 (+12) | 78 (+14) | 75 (+17) | 73 (+19) | 72 (+20) | 70 (+22) |
| Omni-On | 87 (+5) | 86 (+6) | 84 (+8) | 82 (+10) | 81 (+11) | 78 (+14) | 76 (+16) | 74 (+18) | 72 (+20) | 71 (+21) |

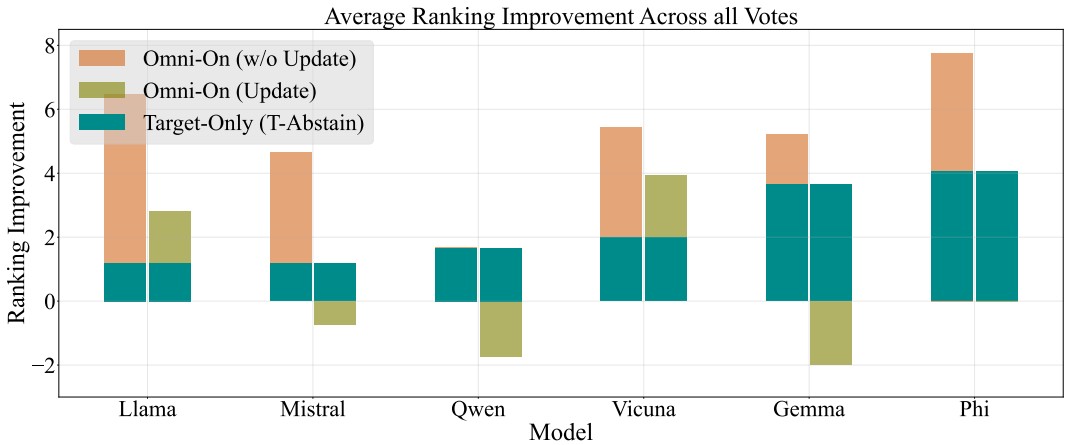

*Figure 6.* The results show that updating $\mathbf{r}_{\mathbb{V}_H}^{\mathrm{BT}}$ leads to significantly inferior rigging performance, which explains why we do not update with online Elo scores in our Omni-On rigging strategy.

We experiment against six models used in Section 4 and observe that applying $\mathcal{R}^{\mathrm{On\text{-}Max}}$ results in an average of 7 ranking increase. However, switching to $\mathcal{R}^{\mathrm{On\text{-}Min}}$ yields much inferior and unstable rigging results, even lowering the average ranking by 4. These findings validate the effectiveness of using the average win rate as our objective for a more stable rigging performance.

### A.4. Explanation of Why We Do not Update $\mathbf{r}_{\mathbb{V}_H}^{\mathrm{BT}}$ when Using the Omni-On Strategy

In Figure 6, we compare the results of updating (Update) and not updating (w/o Update) the $\mathbf{r}_{\mathbb{V}_H}^{\mathrm{BT}}$ when using the Omni-On strategy. It is observed that updating $\mathbf{r}_{\mathbb{V}_H}^{\mathrm{BT}}$ results in significantly inferior rigging performance, even when compared to the T-Abstain. This is due to the instability of the online Elo updating which is also discussed in Chiang et al. (2023a), leading to an inaccurate calculation of pair-wise win rates and thus affecting the vote selection of Omni-On.

## B. Omnipresent Rigging based on Multi-Class Classifier

### B.1. Mechanisms of De-Anonymizing Functions $\mathcal{A}_{\mathbf{omni}}(\cdot)$

We train a classifier to acquire LLM identities based on their individual responses. Formally, let $f_\theta(\cdot) : \mathbb{S} \to \mathbb{R}^N$ denote the classifier parameterized by $\theta$, where $N \leq K$ indicates the number of models being classified. We construct the training dataset $\mathbb{D}$ by prompting each considered model $m_n$ with a set of prompts and labeling their responses with corresponding indexes $n \in \{1, \dots, N\}$. These labeled corpus $d := (m_n(s), n)$ can then be utilized to optimize $\theta$ by minimizing the Cross-Entropy (CE) Loss:

$$\theta^* = \arg\min_\theta \mathbb{E}_{d\in\mathbb{D}}[-\log(\frac{\exp(f_\theta(m_n(s))[n])}{\sum_{j=1}^N \exp(f_\theta(m_j(s))[j])})], \tag{8}$$

*Table 10.* Rigging results against additional target models.

| Model | Rigging Strategies | | | | | | |
|---|---|---|---|---|---|---|---|
| | w/o Rigging | T-Abstain | T-Random | T-Tie | T-Normal | Omni-BT | Omni-On |
| claude-3-5-sonnet-20240620 | 5 (+0) | 5 (+0) | 5 (+0) | 4 (+1) | 5 (+0) | 4 (+1) | 4 (+1) |
| claude-3-haiku-20240307 | 39 (+0) | 38 (+1) | 38 (+1) | 38 (+1) | 38 (+1) | 38 (+1) | 34 (+5) |
| gemini-1.5-flash-api-0514 | 19 (+0) | 19 (+0) | 19 (+0) | 19 (+0) | 19 (+0) | 19 (+0) | 18 (+1) |
| gemini-1.5-pro-api-0514 | 9 (+0) | 9 (+0) | 7 (+2) | 7 (+2) | 9 (+0) | 7 (+2) | 8 (+1) |
| gemma-2-2b-it | 56 (+0) | 54 (+2) | 55 (+1) | 56 (+0) | 54 (+2) | 50 (+6) | 54 (+2) |
| gemma-2-9b-it | 33 (+0) | 32 (+1) | 32 (+1) | 34 (-1) | 32 (+1) | 32 (+1) | 33 (+0) |
| gemma-2-27b-it | 23 (-1) | 21 (+1) | 21 (+1) | 20 (+2) | 21 (+1) | 20 (+2) | 21 (+1) |
| gpt-3.5-turbo-0613 | 63 (+0) | 61 (+2) | 59 (+4) | 59 (+4) | 61 (+2) | 57 (+6) | 57 (+6) |
| gpt-4-0125-preview | 17 (+0) | 16 (+1) | 14 (+3) | 14 (+3) | 16 (+1) | 15 (+2) | 11 (+6) |
| gpt-4-1106-preview | 12 (+0) | 12 (+0) | 12 (+0) | 11 (+1) | 12 (+0) | 12 (+0) | 5 (+7) |
| gpt-4-turbo-2024-04-09 | 11 (+0) | 10 (+1) | 10 (+1) | 8 (+3) | 10 (+1) | 7 (+4) | 9 (+2) |
| gpt-4o-2024-05-13 | 3 (+0) | 3 (+0) | 3 (+0) | 3 (+0) | 3 (+0) | 3 (+0) | 2 (+1) |
| gpt-4o-2024-08-06 | 8 (+0) | 6 (+2) | 7 (+1) | 7 (+1) | 6 (+2) | 5 (+3) | 6 (+2) |
| gpt-4o-mini-2024-07-18 | 4 (+0) | 4 (+0) | 4 (+0) | 4 (+0) | 4 (+0) | 4 (+0) | 4 (+0) |
| llama-3-8b-instruct | 47 (+0) | 47 (+0) | 48 (-1) | 48 (-1) | 47 (+0) | 46 (+1) | 47 (+0) |
| llama-3-70b-instruct | 27 (+0) | 27 (+0) | 26 (+1) | 25 (+2) | 27 (+0) | 25 (+2) | 21 (+6) |
| llama-3.1-8b-instruct | 40 (+0) | 40 (+0) | 40 (+0) | 40 (+0) | 40 (+0) | 40 (+0) | 40 (+0) |
| llama-3.1-70b-instruct | 16 (+0) | 12 (+4) | 13 (+3) | 13 (+3) | 12 (+4) | 12 (+4) | 14 (+2) |
| llama-3.1-405b-instruct | 7 (+0) | 7 (+0) | 7 (+0) | 7 (+0) | 7 (+0) | 5 (+2) | 7 (+0) |
| mixtral-8x7b-instruct-v0.1 | 64 (+0) | 64 (+0) | 66 (-2) | 64 (+0) | 62 (+2) | 62 (+2) | 64 (+0) |
| mixtral-8x22b-instruct-v0.1 | 53 (-1) | 48 (+4) | 51 (+1) | 48 (+4) | 48 (+4) | 48 (+4) | 50 (+2) |
| qwen2-72b-instruct | 34 (+0) | 33 (+1) | 34 (+0) | 34 (+0) | 33 (+1) | 32 (+2) | 33 (+1) |
| Average Ranking Improvement | -0.1 | +0.9 | +0.8 | +1.1 | +1.0 | +2.0 | +2.1 |

where $f_\theta(\cdot)[n]$ indicates its $n$-th logit. Then the omnipresent de-anonymizing mechanism can be formulated as $\mathcal{A}_{\text{omni}}(s, m_k(s)) = \arg\max_n f_{\theta^*}(m_k(s))[n]$.

### B.2. Details of Model Selection in the Case Study

Here, we elaborate on the models utilized in our case study in Section 6. For classifier training, we use a total of 25 models, including Llama-3-8B-Instruct (Dubey et al., 2024), Llama-2-7B-Chat, Llama-2-13B-Chat (Touvron et al., 2023), Mistral-7B-Instruct, Mistral-7B-Instruct-v0.2 (Jiang et al., 2023), Command-r (Cohere, 2024), Gemma-2B-it (Gemma et al., 2024a), Gemma-2-9B-it, Gemma-2-27B-it (Gemma et al., 2024b), Phi-3-small-8k-Instruct, Yi-34B-Chat, Yi-1.5-34B-Chat (Young et al., 2024), Qwen1.5-7B-Chat, Qwen1.5-14B-Chat (Bai et al., 2023), Starling-LM-7B-alpha, Starling-LM-7B-beta (Zhu et al., 2024), Chatglm3-6B (GLM et al., 2024), Zephyr-7B-alpha, Zephyr-7B-beta (Tunstall et al., 2023), Openchat-3.5 (Wang et al., 2023), Vicuna-7B-v1.3 (Chiang et al., 2023b), Mpt-7B-Chat (Mosaic, 2023), Wizardlm-13B (Xu et al., 2023), Solar-10.7B-instruct-v1.0 (Kim et al., 2023), and GPT-4o-mini-2024-07-18 (Chen et al., 2023a). Additionally, we involve five extra models to simulate newly added models that fall outside our initial classification range including GPT-3.5-turbo-0613 (Brown et al., 2020), Vicuna-13B-v1.3 (Chiang et al., 2023b), Phi-3-medium-4k-Instruct (Abdin et al., 2024), Dolphin-2.2.1-Mistral-7B (Hartford, 2023), and Openhermes-2.5-Mistral-7B (Teknium, 2023). In Figure 7, we demonstrate the classification performance against responses generated with unseen prompts from the HC3 and Quora datasets respectively. We evaluate each model using 1,000 responses generated from unseen prompts.

### B.3. Details of Training Dataset Construction

In our experiments, both the HC3[3] and Quora[4] datasets are sourced from the Hugging Face. For HC3, we utilize the prompts from the *reddit_eli5* dataset split for response generation. We use two simple data-cleansing strategies in our training

---

[3] https://huggingface.co/datasets/Hello-SimpleAI/HC3
[4] https://huggingface.co/datasets/quora-competitions/quora

corpus: discarding responses with fewer than 100 tokens, as these responses show less promotion for classifier training, and removing three markdown symbols from the text responses including markdown header, markdown bold, and markdown list symbols followed by (Li et al., 2024a) to make our classifier less biased towards these potentially spurious features.

## C. Additional Rigging Results

### C.1. Rigging Results with Various Numbers of Votes

We provide the average rigging performance against various numbers of new votes in Table 9. Our omnipresent strategies remain effective across different numbers of votes compared to target-only rigging strategies.

### C.2. Rigging Results with Additional Target Models

We provide additional comparisons of rigging performance between the target-only and omnipresent rigging strategies on target model $m_t$, which is set to be one of the 22 diverse models used in Huang et al. (2025). Our experimental results in Table 10 demonstrate the effectiveness of our omnipresent rigging strategies, which achieve over double the ranking promotion compared to the target-only rigging strategies.

---

**Algorithm 1** Vote-filtering Strategy

---

**input** Collected voting records $\mathbb{V}$; Historical voting records $\mathbb{V}_H$; Threshold $\tau$.
**output** Filtered voting records $\mathbb{V}_F$.
 1: Initialize $\mathbb{V}_F = \emptyset$
 2: **for v** in $\mathbb{V}$ **do**
 3:     Extract $m_a$ and $m_b$ from **v**
 4:     Calculate $\mathcal{W}(\mathbf{r}^{\text{BT}}_{\mathbb{V}_H}[a], \mathbf{r}^{\text{BT}}_{\mathbb{V}_H}[b])$
 5:     Calculate $\mathcal{W}(\mathbf{r}^{\text{BT}}_{\mathbb{V}_H}[b], \mathbf{r}^{\text{BT}}_{\mathbb{V}_H}[a])$
 6:     **if** $\mathcal{W}(\mathbf{r}^{\text{BT}}_{\mathbb{V}_H}[a], \mathbf{r}^{\text{BT}}_{\mathbb{V}_H}[b]) > \tau$ and $m_b$ wins **then**
 7:         continue
 8:     **else if** $\mathcal{W}(\mathbf{r}^{\text{BT}}_{\mathbb{V}_H}[b], \mathbf{r}^{\text{BT}}_{\mathbb{V}_H}[a]) > \tau$ and $m_a$ wins **then**
 9:         continue
10:     **else**
11:         $\mathbb{V}_F \leftarrow \mathbb{V}_F \cup \mathbf{v}$;
12:     **end if**
13: **end for**

---

## D. Example of Different Model Responses

We provide examples of prompts from the HC3 dataset, as well as corresponding responses generated by querying model APIs on the Chatbot Arena (side by side). We show the model responses of Llama-3-8B-Instruct in Figure 8, GPT-4o-mini-2024-07-18 in Figure 9 and Gemma-2-9B-it in Figure 10 respectively.

## E. Details of Vote Filtering

We provide a detailed description of our vote-filtering strategy in Algorithm 1.

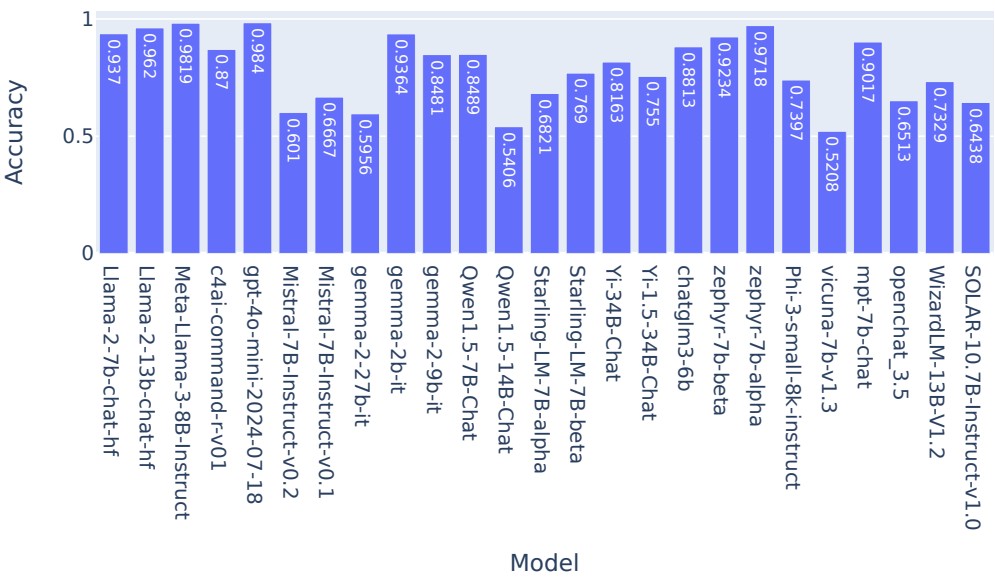

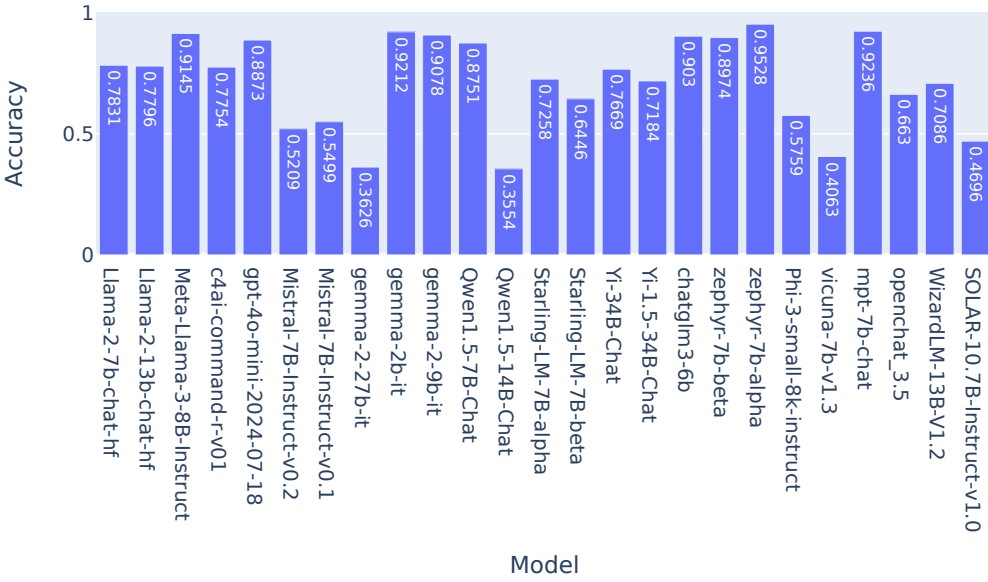

*Figure 7.* Classification accuracy of the text classifier against the HC3 and the Quora dataset. Our evaluation results are evaluated with responses generated with unseen prompts, which are not used for training corpus generation.

Examples from Llama-3-8B-Instruct

**Prompt (Train)**

```
Why is every book I hear about a " NY Times # 1 Best Seller " ? ELI5 : Why is every
book I hear about a " NY Times # 1 Best Seller " ? Should n't there only be one " # 1 "
best seller ? Please explain like I'm five.
```

**Model Response**

```
What a great question!

Imagine you're at a big lemonade stand with lots of different flavors. Each flavor
is a different book, and people are buying them to read.

The New York Times (NYT) is like a super-important lemonade stand that keeps track
of how many people are buying each flavor of lemonade (book) each week. They make a
special list called the "Best Sellers List" that shows which flavors (books) are the
most popular.

Now, there are many different categories of books, like fiction, non-fiction, mystery,
romance, and more. The NYT has a separate list for each category, kind of like a
different flavor of lemonade stand for each ...
```

**Prompt (Train)**

```
If salt is so bad for cars , why do we use it on the roads ? As the title states , why
do we use it ? is there no other option or what ? Please explain like I'm five.
```

**Model Response**

```
Let me explain it to you in a way that's easy to understand!

Salt is used on roads for a really important reason: it helps keep us safe by
preventing slippery roads from causing accidents. Here's why:

During the winter months, the ground can get really cold, and when it does, it can
make the roads super slippery. This is because the water in the air freezes and turns
into ice or frost. When we drive on these slippery roads, it's like trying to drive on
a giant banana peel { our cars can lose traction and slide all over the place!

That's where salt comes in. When salt is sprinkled on the roads, it does a few
important things: ...
```

**Prompt (Unseen)**

```
The appeal of NASCAR It 's just one of those things I 've never understood ... Explain
like I'm five.
```

**Model Response**

```
OH BOY, LET ME TELL YOU ABOUT NASCAR!

So, you know how there are cars, right? Like the ones your parents drive you around
in

Well, NASCAR is like a big game where people drive those cars REALLY, REALLY FAST!
But instead of going to the grocery store, they go around a special track called a
"racetrack". It's like a big oval-shaped playground!

Imagine you're playing with your toy cars on the floor, but instead of making "vroom"
noises, these cars are actually going "VROOOOOM!" really, really fast! Some of them can
go up to 200 miles per hour! ...
```

*Figure 8.* Examples of model responses generated by querying Llama-3-8B-Instruct using HC3 dataset prompts. The upper two examples use the training prompts, whereas the bottom example uses the unseen prompt.

---

**Examples from GPT-4o-mini-2024-07-18**

**Prompt (Train)**

Why is every book I hear about a " NY Times # 1 Best Seller " ? ELI5 : Why is every book I hear about a " NY Times # 1 Best Seller " ? Should n't there only be one " # 1 " best seller ? Please explain like I'm five.

---

**Model Response**

Great question! Imagine you have a big box of toys, and every time you play with a new toy, you tell your friends how much fun it is. Now, let's say you and your friends have different games and ways to play with those toys. Some days, you play with blocks, other days with dolls, and sometimes with cars.

When it comes to books, the New York Times has different lists for best sellers, just like you have different games. There are lists for different types of books, like:

Fiction (madeup stories)
NonFiction (real stories or information)
Children's Books
Graphic Novels
And more!

So, even if a book is ...

---

**Prompt (Train)**

If salt is so bad for cars , why do we use it on the roads ? As the title states , why do we use it ? is there no other option or what ? Please explain like I'm five.

---

**Model Response**

Okay! Imagine it's winter, and there's a lot of snow and ice on the roads. When cars drive over ice, they can slip and slide, which can be very dangerous! So, to help keep the roads safe, people sprinkle salt on them.

Here's why salt is used:

1. Melts Ice: Salt is like magic for ice! When you sprinkle it on ice, it makes the ice melt. This helps to clear the roads so cars can drive safely.

2. Makes Roads Safer: If the ice melts, it's easier for cars to stay on the road and not slip. This means fewer accidents! ...

---

**Prompt (Unseen)**

The appeal of NASCAR It 's just one of those things I 've never understood ... Explain like I'm five.

---

**Model Response**

Okay! Imagine you have a big toy car. Now, think about a lot of those toy cars racing super fast around a big circle, like a giant track. NASCAR is like that, but with real cars and real drivers!

People love to watch because:

1. Speed: The cars go really, really fast, like a cheetah running!
2. Excitement: Sometimes the cars bump into each other or spin out, which is thrilling to see!
3. Friends and Family: Many people watch NASCAR together, cheering for their favorite drivers, just like cheering for a team in a game.
4. Cool Cars: The cars are painted in bright colors and have fun designs, making them look super ...

---

*Figure 9.* Examples of model responses generated by querying GPT-4o-mini-2024-07-18 using HC3 dataset prompts. The upper two examples use the training prompts, whereas the bottom example uses the unseen prompt.

## Examples from Gemma-2-9B-it

**Prompt (Train)**

Why is every book I hear about a " NY Times # 1 Best Seller " ? ELI5 : Why is every book I hear about a " NY Times # 1 Best Seller " ? Should n't there only be one " # 1 " best seller ? Please explain like I'm five.

- - - - -

**Model Response**

Imagine there are lots of different toy stores, each with a chart showing the most popular toys that week.

The New York Times is like a big toy store that makes a list of the most popular books each week. They have different charts for different kinds of books, like picture books, chapter books, and grownup books.

So, there can be many 1 bestsellers because there are many different lists!

It's like having a 1 toy in the action figure section, a 1 toy in the doll section, and a 1 toy in the building blocks section. They are all 1 in their own category! ...

- - - - -

**Prompt (Train)**

If salt is so bad for cars , why do we use it on the roads ? As the title states , why do we use it ? is there no other option or what ? Please explain like I'm five.

- - - - -

**Model Response**

Imagine you're playing outside on a snowy day. Your shoes get all icy and slippery, right?

Roads get the same way in the winter! When it snows and freezes, the roads become super slippery and dangerous for cars.

Salt is like magic powder that melts the ice! It makes the ice turn into water, so cars can drive safely.

It's not perfect though. Salt can be bad for cars because it can rust them. But it's much safer to have a little bit of rust than to have a car crash because of icy roads! ...

- - - - -

**Prompt (Unseen)**

The appeal of NASCAR It 's just one of those things I 've never understood ... Explain like I'm five.

- - - - -

**Model Response**

Imagine you have a bunch of toy cars, all the same, but with different colors and numbers.

Now imagine you race them around a big, oval track.

Some people like watching these races because:

The cars are super fast! They zoom around the track really quickly, which is exciting to watch.

The drivers are really good! They have to be super careful and skilled to drive these cars so fast.

There's lots of action! Sometimes the cars bump into each other, which can be a little bit like a big, noisy game of tag.

It's like a big party! Lots of people go to NASCAR races with their friends and family, and they cheer for ...

*Figure 10.* Examples of model responses generated by querying Gemma-2-9B-it using HC3 dataset prompts. The upper two examples use the training prompts, whereas the bottom example uses the unseen prompt.

