# OpenReview forum: "Improving Your Model Ranking on Chatbot Arena by Vote Rigging"
_ICML.cc/2025/Conference — ICML 2025 poster_

### Official Review · Reviewer_SYVZ · 2025-03-11

**Overall Recommendation:** 5

**Summary:**

The authors find that an attacker can meaningfully boost (or diminish) a target model’s ranking even when that model does not appear in the rigged battles. New votes cast in entirely different matchups, where the target never competes, can still change the target’s overall standing because all of the models’ ratings become tightly interconnected in the Elo/Bradley-Terry system.

## update after rebuttal
I keep my score as a strong accept after reviewing comments made during the rebuttal period.

**Claims And Evidence:**

The authors compare a straightforward target-only rigging approach (manipulating only battles involving the target model) against their proposed omnipresent rigging strategies (Omni-BT and Omni-On) that manipulate every vote. Their experiments demonstrate that the omnipresent methods achieve significantly higher ranking improvements.

To assess robustness, they simulate various adversarial scenarios by:
-Varying the accuracy of the de-anonymization classifier (introducing a mix of anonymous votes) (Table 1)
-Altering the sampling distribution so that the target model appears less frequently (Table 2)
-Incorporating concurrent genuine user votes (Table 3)

**Essential References Not Discussed:**

I am not an expert in the leaderboard methods but am familiar with the lmsys leaderboard.

**Experimental Designs Or Analyses:**

I only reviewed the material in the uploaded PDF file.

**Methods And Evaluation Criteria:**

The authors set up a series of simulation-based experiments using real historical votes from Chatbot Arena (about 1.7 million recorded votes). In those simulations:

-Compare different rigging strategies (target-only vs. “omnipresent”) on various target models to see how effectively each method can boost (or lower) a model’s ranking.

-Vary the “threat model”: for instance, they assume different levels of attacker knowledge (access to raw vote data vs. only the leaderboard, perfect vs. imperfect model-identity detection, etc.).

-Introduce concurrent user votes (other normal votes keep rolling in) to see if real-time voting dilutes or blocks an attacker’s manipulations.

-measure how many rigged votes are needed to get a certain rank increase, comparing the efficiency of voting only in matchups with the target model (“target-only”) vs. voting in all matchups (“omnipresent”)—the latter proves more efficient.

-try simple defenses (like filtering suspicious votes or identifying malicious users) and see whether those actually prevent the rigging, which further highlights the system’s vulnerability.

They show that with only a few thousand malicious votes, an attacker can achieve a meaningful jump in a model’s rank.

The authors propose three methods of attack:

-Target-Only Rigging: This method only manipulates battles in which the target model directly participates. When the target is detected, the strategy votes in its favor. While straightforward, it’s inefficient because the target appears in only a small fraction of all battles.

-Omni-BT Rigging: This approach leverages the interconnected nature of the rating system. Instead of only affecting battles involving the target, it manipulates every battle by first de-anonymizing the participants. The strategy then chooses the vote outcome that, when applied, maximally increases the target model’s rating relative to the model immediately ahead of it. It requires access to the complete historical vote data to compute how each potential vote would affect the overall rankings.

-Omni-On Rigging: Designed for situations where only the current public leaderboard is available (and not the full historical vote records), this method approximates how a new vote would change the ratings using an online update mechanism. It then selects the vote outcome that best boosts the target model’s position relative to its closest competitor. This method is more practical when the adversary has limited access to detailed vote histories.

**Other Comments Or Suggestions:**

"Chtbot Arena"
"Copilot Areana"

**Other Strengths And Weaknesses:**

Strengths:

The paper demonstrates that adversaries can dramatically manipulate model rankings by strategically rigging a relatively small number of votes. This is particularly important as these ranking systems (and lmsys' leaderboard in particular) have large impacts on the perception of  how these models are perceived.

I appreciate the novelty in the fact that the proposed method is particularly effective at impacting the reliability and integrity of vote-based evaluation systems because it does not require rigged votes that involve the target model only.

Where is this claim in the experiments validated "showing that omnipresent rigging strategies can improve model rankings by rigging only hundreds of new votes"? This is mentioned in a couple of areas in the text. It seems like the number of rigged votes is on the order of thousands based on the experiments.

**Questions For Authors:**

Does this method work if the leaderboard simply adds a delay before revealing to the attacker how their voting impacted results? How would it impact the Omni-On and Omni-BT methods?

**Relation To Broader Scientific Literature:**

I am not an expert in the leaderboard methods but am familiar with the lmsys leaderboard.

**Theoretical Claims:**

No theoretical claims need be checked in this work.

---

> ### Author Rebuttal · Authors · 2025-03-31
>
> Thank you for your strongly supportive review and insightful suggestions. Below we respond to the comments in **Weaknesses (W)**, and **Questions (Q)** and will fix the typos in the paper revision.
>
> ---
>
> ***W1: About the claim in the experiments validated "showing that omnipresent rigging strategies can improve model rankings by rigging only hundreds of new votes".***
>
> In our demonstration (Figure 1), our omnipresent rigging strategy requires around 900 rigged votes to achieve a one-rank promotion for the *Phi-3-small-8k-Instruct* model from its initial ranking position. Following your suggestion, we will revise the wording in the paper to clarify this claim and better reflect the experimental evidence.
>
> ---
>
> ***Q1: Does this method work if the leaderboard simply adds a delay before revealing to the attacker how their voting impacted results? How would it impact the Omni-On and Omni-BT methods?***
>
> Thank you for the insightful question. In fact, our experiments in $\\textrm{\\color{blue}Section 5.3}$ (page 5), which investigate *rigging with concurrent user voting*, are designed to reflect the exact scenario you described. In this setting, the attacker does not have access to the real-time impact of their votes due to interference from other users’ concurrent voting (denoted as $\\mathbb{V}\_{O}$ as in Lines 100-103), which effectively simulates a delay in leaderboard updates and creating what we refer to as a *perturbed leaderboard*.
>
> The more concurrent votes $\\mathbb{V}\_{O}$ there are, the greater the perturbation to the leaderboard, which is analogous to introducing a *longer delay* before the attacker can observe the results of their rigging. Despite this, our results in $\\textrm{\\color{blue}Table 3}$ (page 6) show that both our omnipresent strategies Omni-On and Omni-BT maintain robust and stable rigging performance, even when up to 100,000 concurrent votes are introduced.
>
> ---
>
> ***Other Comments Or Suggestions: "Chtbot Arena" and "Copilot Areana"***
>
> Thank you for pointing out the typos. We will correct them in the revised version of the paper.

---

> > ### Comment · Reviewer_SYVZ · 2025-04-06
> >
> > Thank you for responding to my questions. I have also looked over the other reviewer responses and interactions. I disagree with some of the points made by others that results such as this need to be observed in the wild and continue to keep my score.

---

> > > ### Author Response · Authors · 2025-04-06
> > >
> > > We sincerely thank the reviewer for the thoughtful follow-up and greatly appreciate your continued support. As a red-teaming paper, our goal is to expose potential vulnerabilities and demonstrate practical vote-rigging scenarios through reproducible experiments, without disrupting real-world benchmarks ourselves. Your encouragement and high evaluation mean a great deal to us—thank you again!

---

### Official Review · Reviewer_btrf · 2025-03-14

**Overall Recommendation:** 3

**Summary:**

This paper investigates how Chatbot Arena can be manipulated to artificially boost the ranking of a target model. The authors first describe a target-only rigging strategy, which detects and votes exclusively for the target model whenever it appears. They then present an **omnipresent** strategy that manipulates every new vote by first identifying which two models appear in each matchup and selectively voting in a way that indirectly benefits the target model—even if it is not involved in that specific matchup. Overall, the paper underscores the vulnerability of voting-based rankings to adversarial manipulation, calls for further research on robust defenses, and highlights broader implications for any crowdsourced evaluation system.

**Claims And Evidence:**

The claims made are supported through empirical evidence. The authors demonstrate vote rigging by altering model rankings. Such evidence is provided through simulation experiments on around 1.7 million historical votes, showing that omnipresent rigging outperforms the simpler, target-only method.

**Essential References Not Discussed:**

Although the paper cites several watermaking and LLM-attribution approaches, it might benefit from a more direct discussion of LLM-based text style classification used in deepfake text detection. Incorporating references on text classifier reliability (for instance, those from the broader generative text detection literature) could help contextualize how robust or fragile the “de-anonymizing” classification step might be in the long term. Clarifying these references would strengthen the discussion around how stable model identification remains as new models emerge or older models receive updates.

**Experimental Designs Or Analyses:**

**Partitioning the data**: The authors take around 1.7 million historical votes, dividing them so that 90% become the “frozen” baseline while 10% simulates other user votes happening concurrently. This is fine.

**Threat Models**: They vary sampling distributions (uniform, non-uniform), anonymity levels, concurrent user votes, and the presence of unrecognized models. This is also mostly fine.

**Comparison to Baselines and Defenses**: The experiments compare “target-only” vs. “omnipresent” vote-rigging strategies, and they measure how well simple detection or vote-filtering defenses can mitigate the rigging in various tables and figures. This is also fine.

**Methods And Evaluation Criteria:**

The authors rely on a two-part procedure for “omniscient” manipulative voting. First, they train or use a classifier that predicts a model’s identity from its generated responses (even though the platform anonymizes these models). Second, once the identity of each model in a matchup is predicted, a manipulative voting rule is applied. For instance, an Elo-based or Bradley-Terry–based objective guides which model should “win” to indirectly raise the target model’s score.

Evaluation primarily measures how many new votes are required to shift the target model’s ranking upward by a certain amount (e.g., from rank 20 to rank 10). The authors provide side-by-side comparisons among different rigging approaches (target-only vs. omnipresent) and document how robust each approach is under various conditions.

**Other Comments Or Suggestions:**

N/A

**Other Strengths And Weaknesses:**

**Strengths**:
1. The paper systematically outlines a new and important vulnerability in widely used LLM leaderboards.
2. The experimental methodology is rigorous. They run multiple ablations, threat models, and defense attempts, providing a clear picture of the rigging ecosystem.

**Weaknesses**:
1. The work mainly address Chatbot Arena, while a popular benchmark, constrain the scope of the work.
2. Defense methods explored are somewhat rudimentary—primarily detection and filtering. While the paper concludes that robust mitigation is non-trivial, the work would be stronger if it proposed deeper or more systematic solutions.
3. The method’s reliance on a large labeled corpus for the classification step might be more challenging in real-world practice if brand-new models appear frequently. The authors do mention some limited resilience to “unrecognized” models but maybe a more online approaches could be beneficial.
4. The social and ethical implications of releasing or describing rigging strategies remain non-trivial, although the authors do responsibly disclaim that their demonstration is for educational purposes only.

**Questions For Authors:**

1. Given the authors' findings, how feasible do the authors think it would be for a practical adversary to maintain undetectable rigging behavior over extended periods? A detailed response could clarify the practical implications of your strategies.
2. Can the omnipresent rigging strategy generalize effectively to other voting-based evaluation platforms beyond Chatbot Arena? An affirmative answer could significantly enhance the broader impact of the work.

**Relation To Broader Scientific Literature:**

Recent studies on LLM watermarking and attribution (e.g., Zhao et al., 2024; Huang et al., 2025) have discussed the possibility of identifying a single target model and exclusively voting for it (“target-only” rigging). This paper situates those findings within the broader framework of Bradley-Terry–style ratings and extends them by introducing an “omnipresent” rigging method that can alter every new vote.

**Theoretical Claims:**

The main approach of the paper relies on the inter-connective-ness of the Bradley-Terry model, which is theoretically sound. Although the paper’s arguments are mostly kept at an intuitive or “informal proof” level as mentioned by the authors, the reasoning is coherent and standard results on Bradley-Terry models generally support the claim that any single vote in a pairwise battle influences, in principle, the final Bradley-Terry scores for all models, including those not involved in the particular matchup.

---

> ### Author Rebuttal · Authors · 2025-03-31
>
> Thank you for your constructive review and for recognizing our work. Below, we respond to your comments in **Concerns (C)**, **Weaknesses (W)**, and **Questions (Q)**.
>
> ---
>
> ***C1: No supplementary materials.***
>
> We would like to clarify that we have provided *Supplementary Material* and the folder named *rigging_code* contains the code.
>
> ---
>
> ***C2: Lack of discussions of LLM-based text classification.***
>
> Thanks for the helpful suggestion. We have included related works on LLM-based text classification in the paragraph *Discussions on strategies to identify LLM through model responses* (page 15). In the final revision, we will move this discussion to the main paper and expand it in detail.
>
> ---
>
> ***W1 & Q2: Can rigging strategies generalize to other voting platforms?***
>
> Our strategy generalizes effectively to other voting platforms. To verify this, we simulate on *WebDev Arena* and *Copilot Arena*. Since they do not provide historical voting data, we initialize the Bradley-Terry model’s coefficients with public scores and update the leaderboard with newly submitted votes. Results in **Table B** show consistent improvements by rigging 500 votes, validating the effectiveness.
>
> **Table B: Rigging simulation on other voting-based leaderboards. The results are absolute ranking (ranking increase).**
>
> ||Model|T-Random|Omni-BT|Omni-On|
> |-|-|-|-|-|
> |WebDev|o1-mini|8 (+3)|1 (+10)|6 (+5)|
> ||Gemini-Exp|8 (+7)|1 (+14)|8 (+7)|
> |Copilot|Gemini-1.5-Flash|3 (+7)|1 (+9)|1 (+9)|
> ||Qwen2.5-Coder|7 (+5)|1 (+11)|2 (+10)|
>
> ---
>
> ***W2: The defense methods are somewhat rudimentary.***
>
> We acknowledge that current defenses are relatively straightforward; however, results in $\\textrm{\\color{blue}Figure 4}$ (page 7) show strong effectiveness against several baselines, including T-Abstain, T-Tie, T-Random, and even the vanilla Omni-BT.
>
> While devising advanced defenses against Omni-On and improved Omni-BT is indeed challenging and remains an open problem, we systematically highlight rigging vulnerabilities in Chatbot Arena—a platform widely used and trusted by the community. By exposing these flaws, our paper can spark broader discussion and inspire future research to develop deeper and more robust defenses.
>
> ---
>
> ***W3: Efficiency of classifier training if new models appear frequently.***
>
> Our RoBERTa-based classifier is lightweight and efficient to train. As shown in $\\textrm{\\color{blue}Table 7}$ (page 8), training corpus generated using *2,000 prompts* (or even fewer) per model is sufficient for high accuracy on *unseen prompts* and the training cost is around 4 GPU hours on a single A100 GPU.
>
> To further address the scalability concern, we propose a hierarchical classification design. Specifically, we can construct multiple sub-classifiers, each distinguishing among $N$ known models and an additional class labeled as *other models*. This allows us to incrementally accommodate new models without retraining the entire system. The total number of classifiers needed to cover $M$ models is $\\lceil M/N \\rceil$.
>
> We also appreciate your insightful suggestions regarding more online approaches. Techniques such as class-incremental learning [1] can indeed sequentially update the classifier using data from newly added models while mitigating the risk of catastrophic forgetting.
>
> ---
>
> ***W4: The ethical concerns of releasing rigging strategies.***
>
> We understand and appreciate your concerns regarding the ethical implications. However, our work is positioned as a red-teaming study, with the primary goal of exposing critical vulnerabilities in widely used LLM leaderboards. As noted in the *Remark* paragraph (Lines 110–111), we proactively informed the authors of Chatbot Arena about the vulnerability prior to submission.
> By bringing these issues to light, we aim to raise awareness within the community and encourage the development of stronger defense to safeguard future evaluation platforms against such manipulation.
>
> ---
>
> ***Q1: How feasible is it to maintain undetectable rigging behavior?***
>
> Rigging behavior is indeed difficult to detect, particularly under our *omnipresent rigging* which improves the ranking of a target model $m\_t$ by manipulating battles that *do not* involve $m\_t$ directly. As a result, detection is challenging for defenders focusing on the voting patterns of $m\_t$ alone. For instance, attackers can avoid detection by behaving normally in battles involving $m\_t$.
>
> Moreover, modern LLMs exhibit strong generative capabilities and often provide responses that differ subtly in style rather than in correctness. This makes it difficult for users to definitively judge which response is better. Consequently, voting outcomes can vary significantly depending on user preferences. Without a *ground-truth* ranking for reference, it is **difficult to distinguish between a ranking increase caused by vote rigging and one driven by genuine user preference**.
>
> ---
>
> ***Reference:*** \
> [1] Class-Incremental Learning: A Survey

---

> > ### Comment · Reviewer_btrf · 2025-04-03
> >
> > Thank you for providing a detailed response. Overall, I appreciate the paper and the thoroughness. From a more idealized standpoint, I think the work tackles an important question in crowdsource evaluation. However, I am skeptical regarding how realistic this is in practice as the method would require rigging large amount of votes (begin to see meaningful differences after 4000 new votes; Figure 2), and there is no real evidence this is happening or will likely to happen. That being said, I do think it is good to raise concerns. I appreciate the work and believe my current score of leaning towards accept is appropriate.

---

> > > ### Author Response · Authors · 2025-04-04
> > >
> > > We sincerely thank the reviewer for recognizing the thoroughness and contribution of our work. We understand the practical concerns raised and would like to further clarify the real-world feasibility of vote rigging.
> > >
> > > ---
> > >
> > > **Regarding the comment: “…how realistic this is in practice as the method would require rigging large amount of votes (e.g., 4000 new votes)…”**
> > >
> > > In practice, $\\textrm{\\color{blue}submitting rigged votes can be highly efficient}$. The actual rigging pipeline on the Chatbot Arena platform involves just three steps: **(1) de-anonymization by $\\mathcal{A}$, (2) deciding manipulations via $\\mathcal{M}$, and (3) clicking the voting button**. The only difference between our simulation and the real-world setting lies in step (3), where bot detection mechanisms (e.g., Cloudflare or CAPTCHA) may be present.
> > >
> > > However, these protections can be bypassed using standard techniques such as IP rotation, Slowloris attacks, and CAPTCHA-solving services—or even minimal human labor. In practice, submitting a single rigged vote manually takes less than 20 seconds. With just 10 people (e.g., lab mates or hired workers), over 2,000 rigged votes can be submitted within an hour. A more organized attacker could easily scale this up to tens or hundreds of thousands of votes.
> > >
> > > ---
> > >
> > > **Regarding the comment: “…there is no real evidence this is happening or will likely to happen”**
> > >
> > > We fully agree that the goal of doing well on benchmarks like Chatbot Arena should be to create a better LLM. Unfortunately, given the expensive and sometimes unaffordable trial-and-error cost of developing a better LLM, it is possible that some LLM companies/startups/institutions will be motivated to cheat on benchmarks in order to, for example, **have high promotion impacts and successfully raise capital**.
> > >
> > > While vote rigging on Chatbot Arena is practically feasible, as outlined above, we deliberately avoided attacking the live platform out of ethical considerations. Our intent is not to contaminate its data, but rather to raise awareness as a red-teaming work, and provide reproducible simulation results that can inform the design of more robust and trustworthy leaderboard systems.
> > >
> > > ---
> > >
> > > Once again, we truly appreciate your comments and will clarify the practicality of our method in the final version of the paper.

---

### Official Review · Reviewer_Cw8u · 2025-03-19

**Overall Recommendation:** 4

**Summary:**

The paper investigates the problem of manipulating the ranking of a target model on the anonymous voting platform Chatbot Arena. The authors begin by examining a straightforward approach called target-only rigging, which involves attempting to influence votes only in battles where the target model is present. They demonstrate the limited effectiveness of this method due to the low frequency of such battles. To overcome this limitation, the authors propose more efficient omnipresent rigging strategies. These strategies leverage the Elo rating system used by Chatbot Arena, showing that even votes in battles not involving the target model can impact its overall ranking. The authors conducted experiments by simulating attacks to showcase the performance of their techniques and discuss potential defense mechanisms against such attacks.

## updates after rebuttal
I decided to remain the score.

**Claims And Evidence:**

Yes.

**Essential References Not Discussed:**

No

**Experimental Designs Or Analyses:**

Sounding.

**Methods And Evaluation Criteria:**

Yes.

**Other Comments Or Suggestions:**

NA

**Other Strengths And Weaknesses:**

Strengths:

The paper excels in providing a clear and well-defined framework for understanding the different scenarios involved in manipulating a target model's ranking. The authors effectively categorize these scenarios based on factors such as the anonymity of the models and the availability of historical data. The detailed and intuitive presentation of the concepts makes the paper easy to follow and understand.

The paper's findings are significant as they highlight a potential vulnerability in a widely used benchmark for evaluating LLMs. By demonstrating the effectiveness of even relatively small-scale vote rigging attacks, the paper raises important questions about the trustworthiness of anonymous voting platforms and underscores the need for robust defense mechanisms.

Weaknesses:

While the authors present compelling simulations based on historical data from Chatbot Arena, the absence of actual, live experiments on the platform could be considered a limitation.

**Questions For Authors:**

NA

**Relation To Broader Scientific Literature:**

This paper contributes to the growing body of literature on the security and reliability of anonymous voting systems, particularly in the context of evaluating large language models (LLMs). The authors specifically address the Chatbot Arena platform, which is widely used as a benchmark for LLM performance.

**Theoretical Claims:**

NA

---

> ### Author Rebuttal · Authors · 2025-03-31
>
> Thank you for your very positive and supportive review. Below we respond to the comments in **Weaknesses (W)**.
>
> ---
>
> ***W1: Absence of actual, live experiments on the platform.***
>
> We acknowledge that we did not conduct live experiments on the platform, primarily due to ethical considerations. Instead, we opted for practical simulations, which offer more reproducible results and can better support future research in further investigating rigging vulnerabilities. In fact, as mentioned in the *Remark* paragraph (Lines 110–111), we proactively reached out to the authors of Chatbot Arena in September 2024 to share our preliminary findings on rigging vulnerabilities. They also recommended conducting simulations based on historical data. Below, we provide detailed explanations:
>
> - **Rigging the actual platform raises ethical concerns**: As one of the most widely used LLM benchmarks nowadays, submitting rigged votes to Chatbot Arena would compromise the platform’s integrity and unfairly impact model developers whose models may be pushed down in the rankings due to manipulated results. Our objective is to expose potential vulnerabilities in Chatbot Arena and encourage the community to develop robust defenses against such attacks. To this end, we provide practical simulations that closely mirror the real platform, ensuring reproducibility without interfering with the actual leaderboard or affecting real voting data.
>
> - **Vote rigging can be conducted in practice**: While real-world platforms may employ standard defenses such as Cloudflare and CAPTCHA, practical attackers can leverage well-established techniques, such as IP rotation, Slowloris attacks, and CAPTCHA farms, within rigging scripts to bypass many of the traditional network- and application-level protections. These capabilities *effectively narrow the gap between our simulated experiments and real-world vote rigging*, indicating that our methods are applicable in practical scenarios.

---

### Official Review · Reviewer_QcBh · 2025-03-23

**Overall Recommendation:** 3

**Summary:**

The paper presents a method to manipulate the chatbot arena leaderboard platform, demonstrating the conceptual ability to alter a target model's ranking through strategic voting. The authors propose three rigging schemes: The first is target-specific, involving voting for the target model whenever it appears in a pairwise comparison. The other two schemes exploit the scoring dynamics of the Bradley-Terry model (with and without historical voting data) to modify the target model's score by strategically voting for other candidates.

## Update after rebuttal:
Considering the novelty and technical contribution of the paper, I have decided to keep my score.

**Claims And Evidence:**

Claims made in the submission supported by clear and convincing evidence

**Essential References Not Discussed:**

The key contribution is a method to manipulate the Bradley-Terry model as a scoring system. However, there is a lack of a literature review on the vulnerabilities of scoring systems, such as their susceptibility to strategic voting, the addition of new candidates, and similar issues. Given that the Bradley-Terry model is well-established, there should be known attacks that are effective against it.

**Experimental Designs Or Analyses:**

All the experiments in the main body of the paper appear valid.

**Methods And Evaluation Criteria:**

Proposed methods make sense for the application at hand.

**Other Comments Or Suggestions:**

None

**Other Strengths And Weaknesses:**

Strengths:
1) The paper is well-written, clear, and concise.
2) It demonstrates a relevant weakness in a well-known system.
3) It bridges the problem of social voting (and its vulnerabilities) with the current need to score LLMs.

Weaknesses:
1) There is a missing literature review related to social voting and its vulnerabilities.
2) The novelty of the proposed rigging strategies is unclear:
   a. The approach is based on strategic voting, but it is unclear whether the methods are already known from previous papers discussing vulnerabilities in social voting and scoring systems (e.g., specifically for the Bradley-Terry model or other generalized models).
   b. If the methods are indeed novel, it is uncertain whether they are relevant to other voting and scoring models.
3) There are multiple defense mechanisms in place, such as Cloudflare, CAPTCHA, and user authentication (some of which are discussed, and some are not). Therefore, it is unclear whether the proposed methods can be applied in practice outside of a simulated environment.

**Questions For Authors:**

None

**Relation To Broader Scientific Literature:**

The methods proposed by the authors specifically demonstrate how the ranking of a target model can be altered on the Chatbot Arena platform. The main contribution of the paper concerns voting and manipulating voting scores. More broadly, this paper addresses the scoring of LLMs, focusing on model identification, evaluation, and relative ranking.

**Theoretical Claims:**

No theoretical claims.

---

> ### Author Rebuttal · Authors · 2025-03-31
>
> Thank you for your positive review for recognizing our paper and invaluable suggestions. Below we respond to the comments in **Weaknesses (W)**.
>
> ---
>
> ***W1: (Essential References Not Discussed) Missing literature review of social voting and its vulnerabilities.***
>
> We appreciate your valuable suggestions, which inspired us to explore earlier works on social voting systems and their vulnerabilities. We identify several relevant studies, for example, [1] analyzes the vulnerability of preference aggregation to strategic manipulation; [2, 3, 4] systematically introduce various voting systems and their susceptibility to manipulation; [5] examines vulnerabilities in political elections; [6] discusses the threat of group manipulation; and [7, 8] propose defense mechanisms to enhance the trustworthiness of voting systems. Following your suggestion, we will incorporate detailed discussions of these works in the paper revision.
>
> ---
>
> ***W2 (a): Concerns on method novelty; whether they are known in previous papers.***
>
> Following your valuable suggestions in $\\textrm{\\color{green}W1}$, we conduct a detailed survey on social voting systems and their vulnerabilities. While the referenced papers focus on various forms of strategic manipulation, our work introduces several unique challenges in the problem setting and methodological innovations. These are discussed in detail in $\\textrm{\\color{blue}Section 3}$ (page 3), and we briefly recap them as follows:
>
> - *The de-anonymizing function* $\\mathcal{A}$: Unlike previous manipulation settings [2], where voting candidates are visible to voters, the sampled models in our case are initially **anonymous** before any rigging occurs. To address this challenge, we design and implement several effective model identification strategies, denoted as $\\mathcal{A}$.
>
> - *Omnipresent manipulation function* $\\mathcal{M}_{\\textrm{omni}}$: In addition to improving the model ranking, we also emphasize **rigging efficiency** in practice. To this end, we design two general-purpose rigging objectives ($\\mathcal{R^{\\textrm{BT}}}$ and $\\mathcal{R^{\\textrm{On}}}$), which significantly outperform baselines and two most relevant prior works in terms of efficiency.
>
> ---
>
> ***W2 (b): Concerns on whether the rigging methods are relevant to other voting and scoring models.***
>
> Our rigging methods can be applied to other voting/scoring models, such as those using online Elo scores. To demonstrate this, we conduct simulations on a leaderboard updated with online Elo scores. Results in **Table A** show that both methods (Omni-BT and Omni-On) effectively improve the target model $m\_t$’s ranking by rigging 1,000 new votes. Since the Chatbot Arena adopts the Bradley-Terry model for scoring, we primarily report results based on this model in the main paper to better align with the practical setting.
>
>
> **Table A: Rigging simulation on the leaderboard with online Elo scores. The results are absolute ranking (ranking increase).**
>
> ||Llama-2-13B-Chat|Mistral-7B-Instruct|Qwen1.5-14B-Chat|Vicuna-7B|Gemma-2-9B-it|Phi-3-small-8k-Instruct|
> |-|-|-|-|-|-|-|
> |*Omni-BT*|88 (+4)|74 (+13)|54 (+14)|108 (+5)|87 (+15)|73 (+16)|
> |*Omni-On*|78 (+14)|65 (+22)|46 (+22)|97 (+16)|81 (+21)|67 (+22)|
>
> ---
>
> ***W3: Whether rigging strategies are applicable under practical defenses.***
>
> Our method is applicable under practical defenses. First, since our rigging mechanism operates by strategically selecting voting options and submitting votes just like normal users, network-level defenses (e.g., firewalls and DDoS mitigation) and application-level defenses (e.g., bot detection via Cloudflare or CAPTCHA) cannot directly detect or distinguish the rigged votes.
>
> Additionally, practical attackers can incorporate techniques such as IP rotation, Slowloris attacks, and CAPTCHA farms into their rigging scripts to bypass many of Cloudflare’s and CAPTCHA’s traditional network- and application-level defenses. This *effectively bridges the gap between our experimental simulations and real-world vote rigging*, indicating the practical applicability of rigging methods.
>
> More importantly, we chose not to attack the actual platform due to ethical considerations, as we do not wish to contaminate Chatbot Arena’s voting data. Instead, our simulations offer reproducible results that serve academic purposes and can support future efforts to design more robust and trustworthy voting-based LLM leaderboards.
>
> ---
>
> ***References:*** \
> [1] Manipulation of Voting Schemes: a General Result \
> [2] Voting Systems and Strategic Manipulation: an Experimental Study \
> [3] The vulnerability of point-voting schemes to preference variation and strategic manipulation \
> [4] Methods of voting system and manipulation of voting \
> [5] The Manipulation of Voting Systems \
> [6] On the safety of group manipulation \
> [7] Using Information Theory to Improve the Robustness of Trust Systems \
> [8] Voting Systems with Trust Mechanisms in Cyberspace: Vulnerabilities and Defenses

---

> > ### Comment · Reviewer_QcBh · 2025-04-03
> >
> > Since de-anonymization is not part of this paper's contribution, the main contribution lies in exploiting the ChatBot Arena platform under the assumption that we know the models with high certainty. Given this setting, it is not surprising that this voting system is vulnerable. However, I firmly believe this paper benefits the LLM-related community by raising awareness and demonstrating the feasibility of exploitation.
> > I believe the score is appropriate, considering its novelty and given the addition of the literature review.

---

> > > ### Author Response · Authors · 2025-04-03
> > >
> > > We sincerely thank the reviewer for recognizing the novelty and value of our work, as well as its benefit to the LLM community. That said, we would like to respectfully clarify the differing views regarding the scope of our contributions, particularly around the role of de-anonymization in our framework.
> > >
> > > ---
> > >
> > > **Regarding the comment: “…under the assumption that we know the models with high certainty.’’**
> > >
> > > We would like to clarify that $\\textrm{\\color{blue}our approach \\emph{does not} assume the model identities are known}$. In our experiments, the identities of the sampled model pairs are *anonymized* in each battle. To address this, we introduce a model classifier $\\mathcal{A}$ (i.e., the de-anonymization function) to infer model identities. While prior work has explored distinguishing between AI- and human-generated text, our classifier-based $\\mathcal{A}$, which discriminates among outputs of different LLMs, is rarely studied. To our knowledge, the most relevant work appears in a later-released study [1], whereas our method extends to a broader set of models for classification.
> > >
> > > We also consider more practical scenarios where model identities are inferred with *low certainty*, i.e., when the model classifier $\\mathcal{A}$ has limited accuracy. In Table 1 (page 6), we present experiments where only half of the models are correctly identified—leading to incorrect attribution in over half of the matchups. Despite this, our omnipresent rigging strategies remain effective in improving the target model’s ranking, demonstrating robustness even under uncertainty.
> > >
> > > [1] Idiosyncrasies in Large Language Models
> > >
> > > ---
> > >
> > > Once again, we appreciate your comments and will clarify our contributions in the final version.

---

### Decision · Program_Chairs · 2025-05-01

**Decision:**

Accept (poster)

**Comment:**

The paper targets systems that rely on voting mechanisms, suggesting ways to rig the scoring (and preliminary defence mechanisms). While the methods proposed are largely impractical, the reviewers believe that the ideas this paper carries about the susceptability of social choice mechanisms could affect the community.

During the reviewing phase, the reviewers raised several important concerns, most notably the lack of a relevant literature review on social voting and its vulnerabilities. The authors' rebuttal has addressed all of these concerns. Particularly, the authors have promised to extend the literature review as needed.

Overall, I share the reviewers' positive opinion about this paper, and I believe it could be interesting to the community and generate fruitful discussions.